# A LATS biosensor screen identifies VEGFR as a regulator of the Hippo pathway in angiogenesis

T. Azad[1], H.J. Janse van Rensburg[1], E.D. Lightbody[1], B. Neveu[2], A. Champagne [2], A. Ghaffari[1], V.R. Kay[3], Y. Hao[1], H. Shen[4], B. Yeung[1], B.A. Croy [3], K.L. Guan [5], F. Pouliot[2], J. Zhang[4], C.J.B. Nicol[1] & X. Yang[1]

The Hippo pathway is a central regulator of tissue development and homeostasis, and has been reported to have a role during vascular development. Here we develop a bioluminescence-based biosensor that monitors the activity of the Hippo core component LATS kinase. Using this biosensor and a library of small molecule kinase inhibitors, we perform a screen for kinases modulating LATS activity and identify VEGFR as an upstream regulator of the Hippo pathway. We find that VEGFR activation by VEGF triggers PI3K/MAPK signaling, which subsequently inhibits LATS and activates the Hippo effectors YAP and TAZ. We further show that the Hippo pathway is a critical mediator of VEGF-induced angiogenesis and tumor vasculogenic mimicry. Thus, our work offers a biosensor tool for the study of the Hippo pathway and suggests a role for Hippo signaling in regulating blood vessel formation in physiological and pathological settings.

[1] Department of Pathology and Molecular Medicine, Queen's University, Kingston, Ontario, Canada K7L3N6. [2] Department of Surgery (Urology), CHU de Québec-Université Laval Research Center, Laval University, Quebec, Ontario, Canada G1V 0A6. [3] Department of Biomedical and Molecular Sciences, Queen's University, Kingston, Ontario, Canada K7L 3N6. [4] Department of Cancer Genetics and Genomics, Roswell Park Cancer Institute, Buffalo, NY 14263, USA. [5] Department of Pharmacology, University of California at San Diego, La Jolla, CA 92093, USA. Correspondence and requests for materials should be addressed to X.Y. (email: yangx@queensu.ca)

T he Hippo pathway was originally identified in *Drosophila* and later in mammals[1–7]. Since its discovery, studies have found that Hippo signaling has important roles in development and disease by controlling organ size, maintaining tissue homeostasis/regeneration, and directing stem cell differentiation/ renewal, as well as by promoting tumorigenesis, drug resistance,

and metastasis[8–13]. Dysregulation of the Hippo pathway is frequently observed in human cancers[14,15]. When Hippo signaling is activated by upstream regulators, MST1/2 serine/threonine (S/T) kinases (mammalian homologues of *Drosophila* Hippo) phosphorylate/activate LATS1/2 kinases, which subsequently phosphorylate/inactivate their downstream effectors, transcriptional

co-activator Yes-associated protein (YAP), and its paralogue Transcriptional co-activator with PDZ-binding motif (TAZ). S127-phosphorylated YAP (YAP-pS127) or S89-phosphorylated TAZ (TAZ-pS89) bind to cytoplasmic protein 14-3-3 and are prevented from transactivating downstream gene targets in the nucleus (e.g., *CTGF*, *CYR61*, *FGF1*, etc.)[13]. Although a few regulatory factors/cellular processes acting on the Hippo pathway have been uncovered (actin dynamics, cell matrix stiffness, cell–cell contact, and lysophosphatidic acid (LPA))[16], comprehensive regulator screens have been technically limited. Until now, an absence of available tools precluded measuring the dynamics and activity of the Hippo pathway core components in a quantitative, high-throughput, and non-invasive manner. Thus, how the Hippo pathway is regulated by extracellular stimuli remains incompletely understood.

Vascular endothelial growth factors (VEGFs) are molecules that regulate vasculogenesis during embryogenesis and also angiogenesis in adult tissues during wound healing, inflammation, and tumor growth and metastasis[17–19]. Aberrant signaling from VEGFs and their receptors (VEGFRs) is implicated in human diseases and conditions including cancer, diabetes, hemophilia, miscarriage, atherosclerosis, and age-related macular degeneration (AMD)[17,18,20–22]. Therapeutic strategies targeting VEGF and VEGFR are used to treat multiple diseases[17,23,24]. There is emerging evidence that the Hippo pathway has a role in vascular cell migration and VEGFR-induced developmental angiogenesis[25–27]. However, the precise signaling underlying this interaction and relative importance of Hippo signaling as a mediator of VEGFR function in disease is largely unknown. Indeed, a better understanding of the mechanisms underlying VEGF/VEGFR signaling is critical to fully appreciate and predictably modify their functions in physiology and pathology.

In order to identify upstream regulators of the Hippo pathway, here we develop a bioluminescence-based biosensor to measure the activity of LATS—the central factor of the Hippo signaling pathway[16,28]. We demonstrate that our LATS biosensor (LATS-BS) non-invasively monitors LATS activity in vitro and in vivo in real-time with accurate quantification, high sensitivity, and excellent reproducibility[29]. We perform a screen for upstream kinases that modulate LATS kinase activity using a library of small molecule kinase inhibitors and identify VEGFR signaling as a regulator of the Hippo pathway. We show that signaling between VEGFR, YAP, and TAZ has a critical role in VEGF-induced angiogenesis. Therefore, we have created a new tool to study Hippo pathway function and identify VEGF signaling as a

regulator of the Hippo pathway in angiogenesis and vasculogenic mimicry (VM).

## Results

**Establishment of a LATS-BS.** As LATS phosphorylates S127 on YAP and 14-3-3 binds specifically to phosphorylated but not unphosphorylated S127-YAP[13], we constructed a biosensor that monitors LATS kinase activity by measuring the interaction between pS127-YAP and 14-3-3. Our LATS-BS consists of a minimal YAP fragment that interacts with 14-3-3 in a phosphorylation-dependent manner. We chose not to use the full-length YAP protein, to avoid confounding signals by post-translational modifications of YAP (e.g., phosphorylation at other sites and ubiquitination) by other upstream regulators. We tested whether 15 amino acids of YAP (YAP15) surrounding the S127 LATS phosphorylation site (aa. 120–134; Fig. 1a) could interact with 14-3-3 after phosphorylation by LATS2 kinase. Using in vitro glutathione S-transferase (GST)-pulldown assays, we show that similar to full-length YAP-GST, YAP15-GST could directly bind to 14-3-3 after LATS phosphorylation, whereas a phosphorylation-mutant YAP15-S127A-GST could not (Fig. 1b). We then fused YAP15 and 14-3-3 with N-terminal and C-terminal luciferase fragments (Nluc and Cluc), respectively, to create the LATS-BS (Fig. 1c). As depicted in Fig. 1d and demonstrated in Fig. 1e, cells transfected with LATS-BS alone had low luciferase activity and this was correlated with a low degree of Nluc-YAP15-S127 phosphorylation. Co-transfection of LATS-BS with MST2 was associated with increases in both Nluc-YAP-S127 phosphorylation and luciferase activity, and this effect was suppressed with LPA, an inhibitor of the Hippo pathway. LATS-BS co-expression with both MST2 and LATS2 was correlated with further increases in Nluc-YAP15-S127 phosphorylation and luciferase activity. Collectively, these observations are consistent with a model where MST2 activates LATS2, which phosphorylates Nluc-YAP15-S127, leading to binding with Cluc-14-3-3 and reconstitution of active luciferase (Fig. 1e).

To further validate this model, we mutated conserved residues within the LATS consensus phosphorylation motif (HxRxxS/T) on Nluc-YAP15-S127 to alanine (H122A, R124A, and S127A). Each individual mutation completely abolished Nluc-YAP15-S127 phosphorylation and LATS-BS luciferase activity (Fig. 1f). Likewise, the basal LATS-BS signal was reduced by CRISPR-Cas9 knockout of endogenous MST1/2 in HEK293A (50% reduction due to MST-dependent/independent LATS activation) or more

**Fig. 1** Establishment of a split luciferase LATS biosensor (LATS-BS). **a** Schematic diagram of YAP1 structure with LATS phosphorylation site (S127) and surrounding 15 amino acids (YAP15) indicated. **b** YAP15 is sufficient for interaction with 14-3-3. GST-tagged YAP (YAP-GST), YAP15 with wild-type sequence (YAP15-S127-GST), or LATS phosphorylation-mutant YAP15 (YAP15-S127A-GST) was purified. Five micrograms of GST fusion protein was incubated with recombinant LATS kinase. One hundred micrograms of cell lysate from HEK293 transiently expressing 14-3-3-FLAG was added. YAP, YAP15-S127, or YAP15-S127A/14-3-3 binding was assessed by GST pull-down assay, followed by western blotting. **c** Domain structure of the LATS-BS. For Nluc-YAP15, firefly luciferase amino acids 1-416 (Nluc) were fused to the N-terminal of YAP15 (120–134) separated by a glycine/alanine linker (5'-GGAGG-3'). For 14-3-3-Cluc, luciferase amino acids 394–550 (Cluc) were fused to the C-terminal of 14-3-3 separated by a glycine/serine linker (5'-GGSGGGGSGG-3'). **d** Mechanism of action for the LATS-BS. At baseline, there is no interaction between YAP15 and 14-3-3; thus, the LATS-BS shows minimal bioluminescence activity. However, LATS-dependent phosphorylation of YAP15-S127 leads to 14-3-3 binding, luciferase complementation, and high biosensor signal. **e** Validation of LATS-BS activity. LATS-BS was transfected alone or with LATS2 or/and MST2 into HEK293. Biosensor activity or NLuc-YAP15-S127 phosphorylation was determined 48 h after transfection. For lysophosphatidic acid (LPA) treatment, cells were stimulated with 10 μM LPA for 1 h ($n = 3$). **f** LATS-BS responds specifically to LATS kinase activity. Mutation of the LATS kinase consensus motif (HXRXXS/T; H, histidine; R, arginine; S, serine; T, threonine; X, any amino acid) in Nluc-YAP15 abolishes LATS-BS activation and Nluc-YAP15 S127-phosphorylation ($n = 3$). **g** LATS-BS activity is reduced by *MST* or *LATS* knockout. LATS-BS was transfected into CRISPR-Cas9-generated *LATS1/2* or *MST1/2* knockout HEK293A. Biosensor activity was determined 48 h after transfection ($n = 3$). **h** LATS-BS can be stably expressed to detect LATS activity. HEK293A with doxycycline (Dox)-inducible LATS2 overexpression and stable LATS-BS expression were treated with Dox for the indicated times. Biosensor activity and endogenous YAP (YAP-pS127) phosphorylation status and Nluc-YA15P-S127 (Nluc-YAP15-pS127) were determined by luciferase assay and western blotting, respectively ($n = 3$). Data are represented as mean ± SD

dramatically by LATS1/2 knockout (~ 90% reduction) (Fig. 1g)[30]. Finally, inducible expression of LATS2 in HEK293A cells stably expressing LATS-BS shows that the levels of LATS2 are correlated with the level of endogenous YAP-pS127, Nluc-YAP15-pS127, as well as with LATS-BS activity (Fig. 1h). Therefore, we have constructed a LATS-BS that indeed responds to LATS activity.

Using a similar approach, we further constructed two additional biosensors that may also be used to monitor LATS activity (see Supplementary Fig. 1a–d).

**LATS-BS responds to stimuli regulating the Hippo pathway.** We next sought to validate the LATS-BS and explore potential applications for its use. As expected, the LATS-BS is activated by LATS in various cell lines (Fig. 2a). Similarly, the LATS-BS responded to numerous signals reported to modulate Hippo pathway activity, including cell confluency (Fig. 2b), drugs activating Hippo signaling (forskolin, phosphoinositide 3-kinase (PI3K) inhibitor, phosphoinositide-dependent kinase (PDK) inhibitor, forskolin/3-isobutyl-1-methylxanthine (F-IBMX), and 2-deoxy-glucose)[16,31–34], and Hippo signaling inhibitors (LPA, epidermal

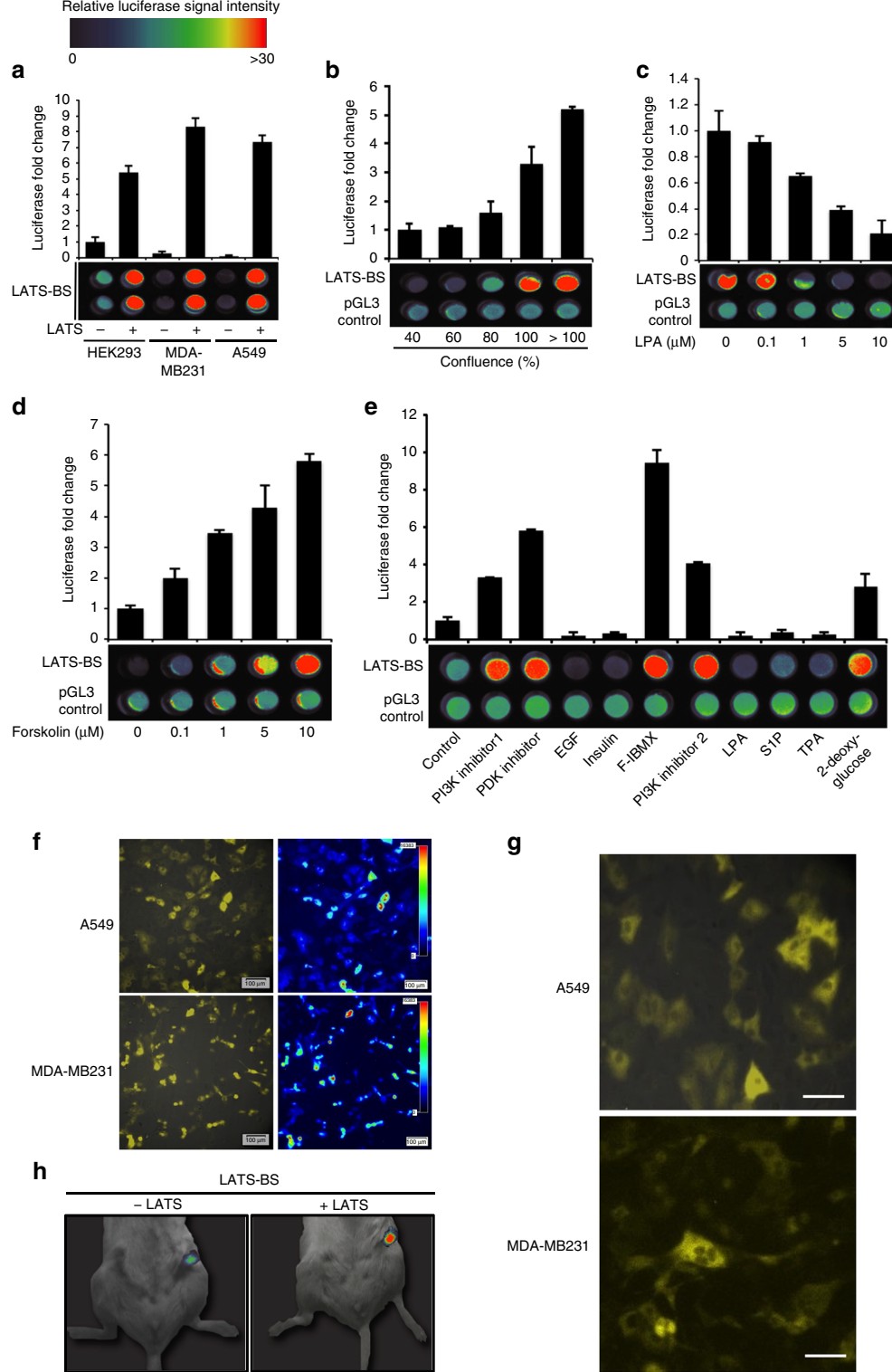

frowth factor (EGF), insulin, sphingosine1-phosphophate (S1P), and 12-O-tetradecanoylphorbol-13-acetate (TPA)) (Fig. 2c–e)[16,30–33]. Notably, for each of these experiments we measured biosensor activity in both cell lysates and in live cells using luciferase assay and bioluminescent imaging (BLI), respectively.

Using the LATS-BS, we also were able to visualize and quantify LATS activity at the individual cell level in living cancer cells using the LV200 BLI system (Fig. 2f and Supplementary Fig. 2a)[35]. In these images, the difference in biosensor signal intensity among individual cells represents altered endogenous LATS activity rather than differential LATS-BS expression levels, as only the LATS-phosphorylated fraction of LATS-BS emits bioluminescence. In addition, both the levels and subcellular localization of LATS-BS bioluminescence (cytoplasm where LATS is expressed) and green fluorescent protein (GFP) (control, nucleus/cytoplasm) were different within the same cell (Fig. 2f, g and Supplementary Fig. 2b). This technology also allowed us to compare the heterogeneity of LATS activity among cancer cell lines by assessing the distribution of luciferase activity (Supplementary Fig. 2c). Finally, using biophotonics BLI, we detected LATS-BS activity in mice, suggesting that this system may be useful for in vivo monitoring of LATS activity (Fig. 2h). Collectively, this data illustrates the broad range of potential applications for the LATS-BS in monitoring Hippo pathway activity.

**Screen for kinases that modulate LATS activity**. We next used our LATS-BS to search for kinases regulating Hippo signaling with a small-scale kinase inhibitor screen (Fig. 3a). Of 80 kinase inhibitors screened, 6 kinase inhibitors activated the biosensor (i.e., inhibitors of VEGFR, MEK, GSK-3, PKB/Akt, EGF receptor (EGFR), and CDK), whereas 6 inhibitors reduced the biosensor signal (i.e., inhibitors of TrkA, Broad, SYK, ATR/ATM, CHK1, and SGK) (Fig. 3b, Table 1). Several candidate LATS regulators identified by our screen were already described in the literature (i.e., EGFR and insulin)[34,36,37], whereas others were novel. We confirmed the results of the screen using our LATS-BS and an STBS-luciferase reporter that measures the transactivating function of YAP/TAZ[38]. Indeed, multiple kinase inhibitors had opposite effects on the LATS-BS and STBS reporter (Fig. 3c,d), suggesting that our screen had identified true regulators of Hippo signaling.

**VEGFR regulates the Hippo pathway**. In our screen, the VEGFR inhibitor SU4312 showed the most dramatic effect on LATS and YAP/TAZ activities (Fig. 3c,d). Thus, we chose to further confirm the interaction between VEGFR and the Hippo pathway. Multiple VEGFR inhibitors activated LATS and inhibited YAP/TAZ (Fig. 4a, b). The messenger RNA levels of CYR61 and CTGF, two downstream transcriptional targets of YAP/TAZ[39], were also

significantly reduced after VEGFR inhibitor treatments (Fig. 4c). Overexpression of either VEGFR1 or VEGFR2 alongside YAP in HEK293 cells inhibited S127 phosphorylation on YAP (Fig. 4d). As a control, the LATS-BS showed no evidence of tyrosine phosphorylation by VEGFR2 (Supplementary Fig. 1e).

The relationship between VEGFR and Hippo signaling could also be modulated by stimulation of VEGFR with its ligand. VEGF treatment suppressed LATS-BS activity in MCF10A cells stably overexpressing VEGFR1 or VEGFR2 (Fig. 4e), whereas LATS inhibition is reversed by the VEGFR inhibitor Axitinib (Fig. 4e), a more specific VEGFR inhibitor than SU4312. Accordingly, VEGF treatment enhanced endogenous CYR61 mRNA expression in a VEGFR-dependent manner in MCF10A-VEGFR1/2, MDA-MB231 breast cancer, human umbilical vein endothelial cell (HUVEC), an immortalized human endothelial cell line Telo-HEC, and human blood outgrowth endothelial cells (BOEC) (Fig. 4f, g and Supplementary Fig. 4a, b). Finally, as LATS inhibits YAP by phosphorylating and sequestering YAP/TAZ in the cytoplasm[3,4], we examined the subcellular localization of YAP or TAZ after VEGF treatment in MCF10A-VEGFR2 (YAP high), MDA-MB231 (TAZ-high), or BOEC (TAZ-high) cells[3,4]. YAP or TAZ are translocated into the nucleus within 15–30 min of VEGF treatment (Fig. 4h–m and Supplementary Fig. 3a–d), suggesting that VEGF/VEGFR activates YAP/TAZ through enhancement of their nuclear localization by inhibiting LATS.

Previous studies have suggested that VEGFR activates PI3K and mitogen-activated protein kinase (MAPK) signaling pathways[40]. Furthermore, others have shown that PI3K and MAPK signaling regulates the activity of Hippo pathway components[34,36,37]. Therefore, we next explored whether VEGFR suppresses LATS and activates YAP/TAZ through MAPK and PI3K signaling. We transfected the LATS-BS or STBS reporter alone or together with VEGFR2 into HEK293A cells and treated the cells with inhibitors of VEGFR, PI3K, AKT/PKB, or MEK. Similar to the VEGFR inhibitor, PI3K, AKT, and MEK inhibitors all blocked both VEGFR2-induced inhibition of LATS (Fig. 4n) and activation of YAP/TAZ (Fig. 4o). Furthermore, as previous studies indicate that VEGF activates downstream signaling through PI3K and MAPK, and that PI3K-Akt regulate the Hippo pathway through MST1, we assessed how VEGF treatment affects phosphorylation of each of the components (e.g., VEGFR, PI3K, ERK1/2, and MST1/2) in the signaling between VEGF and LATS. As shown in Fig. 4p, VEGF treatment enhanced phosphorylation of VEGFR, AKT, and ERK. VEGF also reduced phosphorylation of MST1 and LATS1 (at the MST1 phosphorylation site T1079). These changes in phosphorylation could be reversed by the VEGFR inhibitor axitinib. Together, these data provide strong

**Fig. 2** LATS-BS can be used to measure LATS activity in vitro and in vivo. **a** LATS-BS can be used to assess LATS activity in different cell lines. LATS-BS was transfected alone or together with LATS2-FLAG into HEK293, MDA-MB231, or A549 cells. Biosensor activity was measured by luciferase assay or BLI of live cells ($n = 3$). For BLI, data were heat-mapped (red signal denotes an increase in LATS-BS activity, whereas blue indicates a decrease). **b** The LATS-BS responds to cell confluency. LATS-BS or pGL3-control were transfected into HEK293 at different confluencies. After 48 h, biosensor activity was determined by luciferase assay using cell lysate (top) or bioluminescent imaging (BLI) of live cells (bottom) ($n = 3$). **c**, **d** LATS-BS is inhibited by LPA (**c**) and activated by forskolin (**d**). LATS-BS or pGL3-control were transfected into HEK293. Cells were treated with increasing concentrations of LPA for 1 h or with forskolin for 30 min before luciferase assay and BLI ($n = 3$). **e** LATS-BS responds to drug treatments regulating Hippo signaling. HEK293 were transfected with LATS-BS and treated with the following: PI3K inhibitor 1 (GDC0941), 10 μM for 4 h; PDK inhibitor (GSK2334470), 10 μM for 4 h; EGF, 100 ng ml⁻¹ for 1 h; Insulin, 10 μg ml⁻¹ for 1 h; F/IBMX (Forskolin/IBMX), 10 μM Forskolin/100 μM IBMX for 1 h; PI3K inhibitor 2 (LY294002), 10 μM for 4 h; LPA, 10 μM for 1 h; Sphin gosine1-phosphate (S1P), 1 μM for 1 h; 12-O-tetradecanoylphorbol-13-acetate (TPA), 5 nM for 1 h; 2-deoxy glucose, 25 mM for 1 h. Biosensor activity was determined by luciferase assay or BLI of live cells ($n = 3$). **f**, **g** LATS-BS can be used to observe LATS activity under the microscope. LATS-BS was stably overexpressed in A549 and MDA-MB231. Cells were imaged using LV200 BLI. Images in **g** are higher magnification of images in **f**. Scale bar represents 100 μm (**f**) or 20 μm (**g**). **h** LATS-BS can be used to determine LATS activity in vivo. HEK293 were transfected with LATS-BS. Cells were injected into the mammary fat pad of immunocompromised mice. After 48 h, BLI was performed. Data are represented as mean ± SD

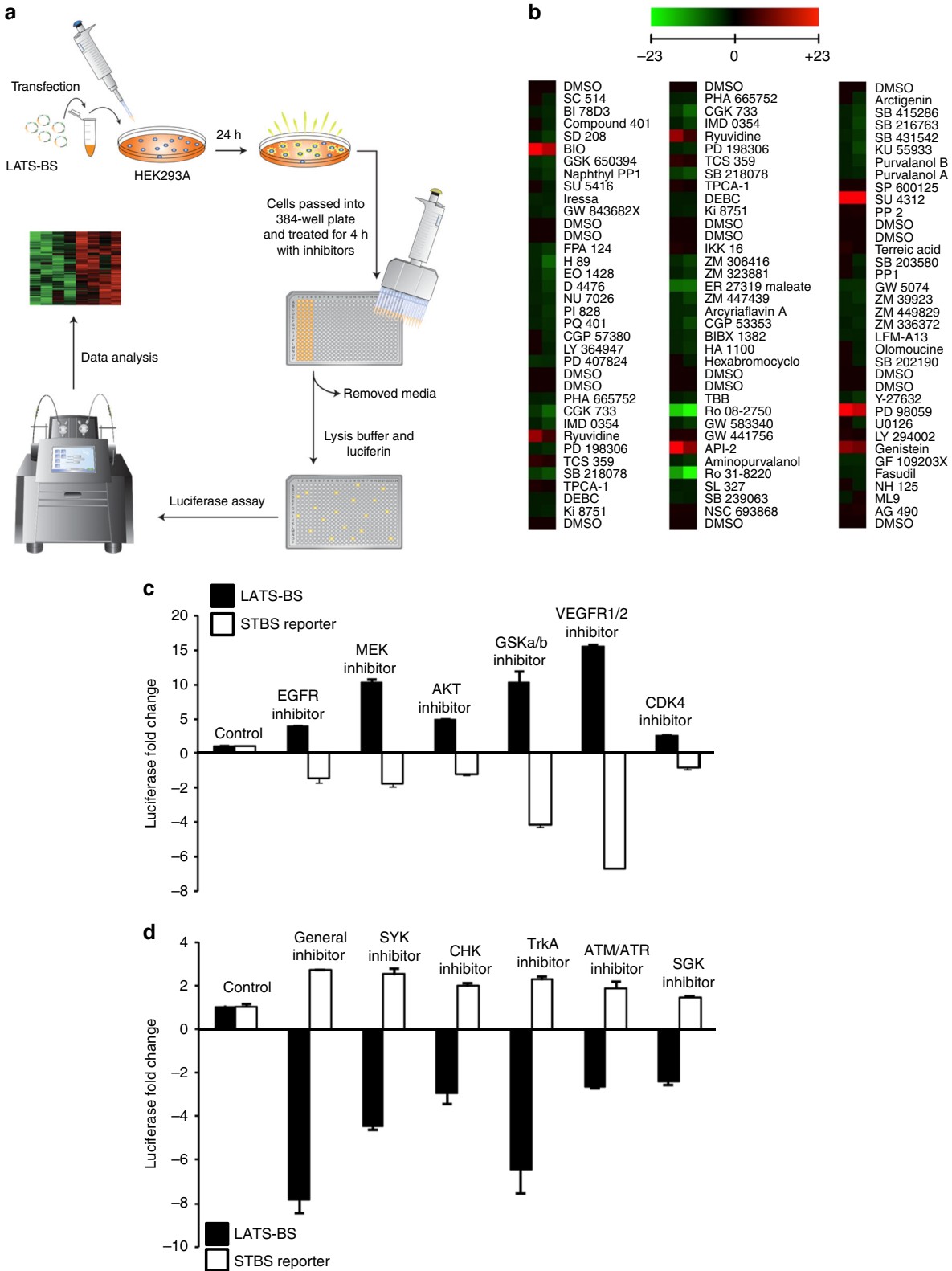

**Fig. 3** Kinase inhibitor screen using the LATS-BS. **a** Experimental design for kinase inhibitor screen. LATS-BS was transfected into HEK293A and cells were passed into a 384-well plate the next day. Forty-eight hours after transfection, cells were treated with a kinase inhibitor library (Tocriscreen Kinase Inhibitor Toolbox) with each drug administered at 10 μM for 4 h in duplicate. Biosensor activity was measured by luciferase assay. **b** Heat map summarizing the results of the kinase inhibitor screen. Red color denotes drug treatments that activated the LATS-BS, whereas green color indicates treatments that inhibited the LATS-BS. **c**, **d** Validation of the top candidate drugs activating (**c**) or inhibiting (**d**) the LATS-BS from the small-scale kinase inhibitor screen. LATS-BS or STBS-luciferase reporter (YAP/TAZ/TEAD reporter) were transfected into HEK293A. Forty-eight hours after transfection, cells were treated with each inhibitor for 4 h at 10 μM. Reporter activity was measured by luciferase assay ($n = 3$). Data are represented as mean ± SD

| Table 1 Results of kinase inhibitor screen using the LATS-BS | | |
| --- | --- | --- |
| **Kinase inhibitor** | **Target kinase** | **Increased fold change** |
| SU 4312 | VEGFR1/2 | 27.88 |
| PD 98059 | MEK | 16.9 |
| BIO | GSKa/b | 14.38 |
| API-2 | PKB | 10.57 |
| Genistein | EGFR | 6.12 |
| Ryuvidine | CDK4 | 5.88 |
| | | |
| **Kinase inhibitor** | **Target kinase** | **Suppressed fold change** |
| Ro 08-2750 | TrkA | 6.51 |
| Ro 31-8220 | General | 5.21 |
| ER 27319 | SYK | 3.71 |
| CGK 733 | ATM/ATR | 1.85 |
| PD 407824 | CHK1 | 1.73 |
| GSK650394 | SGK | 1.73 |

The top kinase inhibitors that activated or inhibited the LATS-BS are shown. The primary target kinases (at 10 μM dose) and fold change in biosensor activity are described for each inhibitor

evidence that VEGF/VEGFR may regulate the Hippo pathway by inhibiting MST1/2 through PI3K/MAPK.

**YAP/TAZ are critical for VEGF-induced angiogenesis in vivo**. VEGF and VEGFR2 are crucial factors in angiogenesis[40]. Likewise, there is emerging evidence linking LATS2 depletion or YAP overexpression to angiogenesis[41–43]. Therefore, we next investigated whether Hippo signaling contributes to VEGF–VEGFR-mediated angiogenesis. We used short interference RNAs (siRNAs) targeting *YAP* and/or *TAZ* to knockdown YAP and/or TAZ in MCF10A-VEGFR2 cells (Fig. 5a). Angiogenesis was assessed using an in vitro tube formation assay after stimulating the cells with VEGF. Although minimal tube formation was observed in wild-type MCF10A cells, overexpression of VEGFR2 increased tube formation in vitro (Fig. 5b) and VEGF treatment further increased this. Notably, knockdown of YAP and/or TAZ completely abolished both basal and VEGF-induced tube formation in MCF10A-VEGFR2 (Fig. 5b, c and Supplementary Fig. 4h). Pharmacological inhibition of YAP/TAZ with a low concentration (100 nM) of verteporfin (VP) achieved the same phenotype as genetic knockdown (Fig. 5b, c). We used the same approach to examine the role of YAP and TAZ during angiogenesis using HUVEC and BOEC cells. In these cells, endogenous YAP expression was barely detectable while TAZ was highly expressed (Fig. 5d, g). Interestingly, TAZ knockdown had a more substantial effect on blocking angiogenesis than YAP knockdown in both HUVEC and BOEC cells (Fig. 5d–i and Supplementary Fig. 4g, j). Importantly, we showed that the reduced angiogenesis caused by YAP/TAZ knockdown or VP treatment is not due to cell death (Supplementary Fig. 4f, 5d).

Mounting evidence suggests that tumor cells can form blood vessel-like structures through VM[44]. VM has important roles in tumor metastasis and drug resistance, and, similar to physiological angiogenesis, is regulated by VEGFR[45,46]. Therefore, we used VEGFR-positive MDA-MB231 cells to assess whether the Hippo pathway is involved in VM. MDA-MB231 cells formed tubes when treated with VEGF, whereas knockdown of YAP and/or TAZ significantly decreased VEGF-induced tube formation (Fig. 5j–l and Supplementary Fig. 4i). VP treatment also blocked tube formation by MDA-MB231 (Fig. 5j–l and Supplementary Fig. 4c–e). Therefore, the Hippo pathway contributes to VEGF-induced VM.

Previous reports have proposed YAP regulates angiogenesis by enhancing pro-angiogenic factor ANG-2 expression[41]. Further,

overexpression of the YAP/TAZ downstream target *CYR61* promotes angiogenesis[47]. Therefore, we examined the cellular levels of ANG-2 and CYR61 in our angiogenesis assays to determine how expression of these targets might relate to the angiogenic phenotype. Knockdown of YAP and/or TAZ dramatically reduced expression of ANG-2 and CYR61 in all of the cell lines examined (Fig. 5a, d, g, j). Interestingly, recombinant ANG-2 and CYR61 proteins could partially rescue tube formation by HUVEC cells after transient knockdown of YAP/TAZ or VP treatment, indicating that *ANG-2* and *CYR61* are critical gene targets regulated by YAP and TAZ in angiogenesis (Fig. 5m, n, and Supplementary Fig. 4k, 5b, c). However, given that these proteins could not fully restore angiogenesis in this model, it is very possible that YAP and TAZ also regulate other genes that contribute to this phenotype.

**YAP/TAZ are critical for VEGF-induced angiogenesis in vivo**. We next tested the relationship between VEGFR, YAP, TAZ, and angiogenesis in other models. First, we used an ex vivo rat aorta model where endothelial cells from sections of rat aorta form sprouts into Matrigel when stimulated with VEGF. Pharmacological inhibition of YAP and TAZ using VP decreased endothelial cell sprouting area in this model in a dose-dependent manner (Fig. 6a, b and Supplementary Fig. 5a). We confirmed that the sprouts observed indeed represent endothelial cell growth by staining the aortic sections for an endothelial cell marker, VE-cadherin (Fig. 6c).

Next, we used Matrigel plugs to explore angiogenesis in vivo in mice through two separate experiments. First, we examined angiogenesis by HUVECs grown in Matrigel plugs seeded with VEGF. VEGF stimulated blood vessel formation by HUVECs, whereas genetic knockdown of YAP/TAZ or YAP/TAZ inhibition by VP dramatically reduced VEGF-induced angiogenesis (Fig. 6d). The blood vessels in this model were positive for human CD31, indicating that the new blood vasculature had been established by HUVECs rather than mouse endothelial cells. We further used Matrigel plugs to evaluate whether YAP and TAZ can have a role in angiogenesis by mouse endothelial cells. We assessed blood vessel formation by murine cells in Matrigel plugs seeded with VEGF. Blood vasculature established by mouse endothelial cells were visualized by staining sections for mouse CD31 endothelial cell marker. In this model, systemic administration of VP reduced VEGF-induced blood vessel formation by mouse endothelial cells in a dose-dependent manner (Fig. 6e).

Finally, we explored how YAP and TAZ contribute to physiological angiogenesis using a mouse retina model. Mice at postnatal day 3 and 4 were treated with VEGF and/or VP, and retinal blood vasculature was observed at postnatal day 5. VP treatment reduced both the overall blood vessel density of the retina, as well as the number of filopodia at the vascular front (a marker of ongoing or active angiogenesis) (Fig. 6f–h). Therefore, we conclude that the Hippo pathway acts downstream of VEGFR to regulate angiogenesis in vitro and in vivo (Fig. 7).

**Discussion**
In this work, we establish the first LATS-BS that can monitor LATS kinase activity and intensity of Hippo signaling in vitro and in vivo. We demonstrate how the LATS-BS can be used to quantify Hippo pathway activity in a wide spectrum of applications ranging from luciferase assays to biophotonics BLI. Although bioluminescence is widely used for reporting promoter activity and imaging tumors in mice, few studies have used it to measure protein function at the cellular level and even fewer studies have examined subcellular protein function using bioluminescence microscopy[35]. By applying new LV200 BLI

technology to our LATS-BS, we were able to detect LATS activity in individual live cells. This system will have exciting applications for evaluating heterogeneous dynamics of LATS activity in cell culture as well as for the real-time monitoring of Hippo signaling responses to various drug treatments. Our in vivo work in mice further illustrates how the LATS-BS could potentially be used to

pre-clinically examine the effects of a variety of drugs on LATS activity in mice.

We designed the LATS-BS using 15 amino acids from the YAP sequence so that it preferentially responds to LATS activity and we have confirmed this to be the case in vitro. However, it is possible that the biosensor may provide insights into other molecular mechanisms that regulate the YAP/14-3-3 interaction.

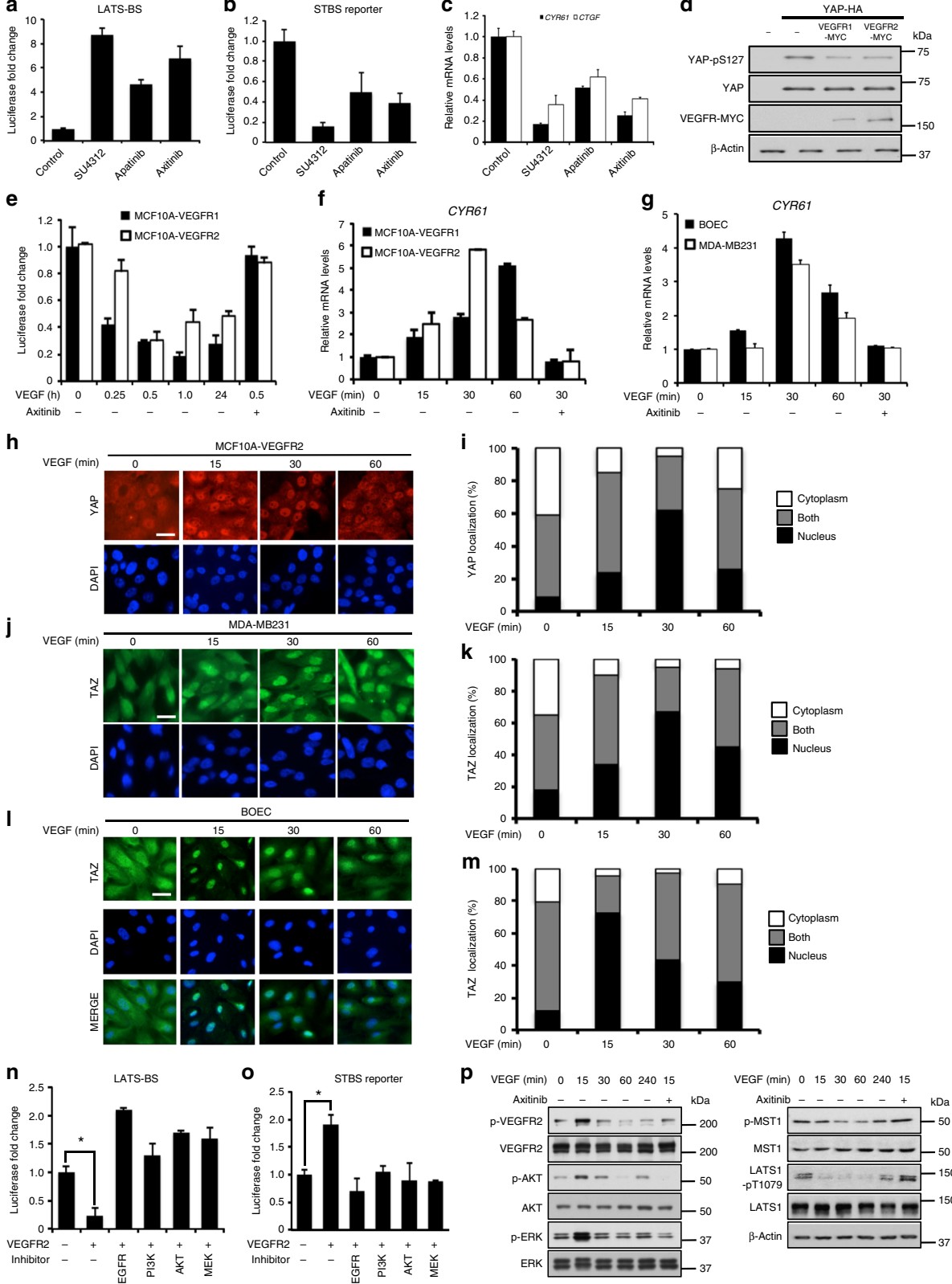

For example, we have previously shown that YAP S127 phosphorylation is inhibited after YAP S128 phosphorylation by NEMO-like kinase under osmotic stress[48]. Thus, the biosensor we have created may have greater utility in determining the activity of additional pathways that affect the YAP/14-3-3 interaction.

In our work, we used the LATS-BS to perform a small-scale screen for regulators of LATS using 80 kinase inhibitors, revealing many kinases acting on the Hippo pathway. We validated VEGFR as a bona fide regulator of the Hippo pathway and established a YAP/TAZ-ANG-2/CYR61 signaling axis that is critical for angiogenesis and VM. Further validation of other candidate regulators of LATS from our screen (e.g., TrkA, SYK, and SGK) may provide insights into Hippo pathway regulation in response to other stimuli and may reveal novel functions for the Hippo signaling. Future large-scale screens using the LATS-BS will undoubtedly identify additional new activators or inhibitors of the Hippo signaling pathway.

We provide evidence that the Hippo pathway is a critical mediator of VEGF/VEGFR-induced angiogenesis and VM. During the revision of our manuscript, additional studies provided evidence that YAP/TAZ are key regulators of vascular cell migration and VEGF-induced developmental angiogenesis[25–27]. Given that VEGF/VEGFR and the Hippo pathway have important roles in cell proliferation and survival, cell migration, as well as stem cell renewal and differentiation[8,10–12,17,18], and that angiogenesis is critical during embryogenesis, organogenesis, and wound-healing[18,40,49], further examination of how these pathways interact in physiological and pathological states is warranted. It will be compelling to further explore how VEGFR-Hippo signaling is involved in these phenomenon.

## Methods

**Plasmid construction and site-directed mutagenesis.** All biosensors were made by overlapping PCR using *Photinus pyralis* firefly luciferase from pGL3 luciferase reporter (Promega) as template. Biosensors were cloned into the BamHI/NotI sites of pcDNA3.1/hygro( + ). Full-length complementary DNAs (cDNAs) of human *VEGFR1* (accession number NM_001159920.1) and *VEGFR2* (accession number NM_002253.2) were subcloned into pcDNA3.1/hygro( + )-myc vector. Full-length cDNAs of human *14-3-3τ* (accession number NM_006826) was subcloned into pcDNA3.1/hygro( + )-3XFLAG vector.

For lentivirus production, *Nluc-YAP15, 14-3-3-Cluc,* or *VEGFR1/2* cDNA were cloned into WPI lentiviral vector. To make GST fusion proteins, oligos coding for YAP15-S127 and YAP15-S127A were annealed and directly cloned into the BamHI/NotI sites of pGEX-4T1. The following primers were used for cloning (restriction enzyme sites underlined):

14-3-3, sense (5′-CTGGATCCATGGAGAAGACTGAGCTGATCC-3′), anti-sense (5′-ATGA AACTGCGGCCGCTTAGTTTTCAGCCCCTTCTGCC-3′); YAP15-S127, sense (5′-GATCCCCACAGCATGTTCGAGCTCATTCC TCTCCAGCTTCTCTGCAGTTGTGAGC-3′), anti-sense (5′- GGCCGCTCA-CAACTGCAGAGAA GCTGGAGAGGAATGAGCTCGAACATGC TGTGG-3′); YAP-S127A, sense (5′-GATCCCCACAGCATGTTCGAGCTCATGCGTC TC

CAGCTTCTCTGCAGTTGTGAGC-3′), anti-sense (5′-GGCCGCTCACAA-CTGCAGAGAAGC TGGAGACGCATGAGCTCGAACATGCTGTGGG-3′); VEGFR1, sense (5′-TCTGGATC CATGGTCAGCTACTGGGACACC-3′) anti-sense (5′-CTACGCGTCTAGATGGGTGGGG TGGAGTAC-3′); VEGFR2, sense (5′-TCTGGATCCATGCAGAGCAAGGTGCTGCTG-3′) (BamHI partial digestion required) anti-sense (5′-TCTACGCGTTTAAACAGGAGGAGAGCT CAGTGTG -3′); Nluc-YAP15, sense (5′-CTGGATCCGCCGCCACCAT-GGAAGACGCCAA AAACATAAAG -3′), anti-sense (5′-ATGAAACTGCGGCCGCTTACAACTGCAGAGAAGCTGG AGAGGAAT-GAGCTCGAACATGCTGTGGGCCTCCAGCTCCTCCTCCA TCCTTGTCAATCAAGGC -3′); 14-3-3-Cluc, sense (5′-CTGGATCCGCCGCCA-CCATGGAGAAGACTGAGCTGATC-3′), sense (5′-GGTAGTGGAGGA-GGAGGTAGTGGTCCTATGATTATGTC C-3′), anti-sense (5′- ACTACCTC-CTCCTCCACTACCTCCTCCTCCGTTTTCAGCCCCTTC TGCCGC-3′), anti-sense (5′-ATGAAACTGCGGCCGCTTACACGGCGATCTTTCC-3′); engineered luciferase LATS-BS, anti-sense (5′-ATGAAACTGCGGCCGCTTACAACT-GCAGAGAAGCTGGAGAGGAATGAG CTCGAACATGCTGTGGCTTCTTG-GCCTTTATGAGGATC -3′); Nluc-YAP15-H122A, anti-sense (5′-ATGAAA-CTGCGGCCGCTTACAACTGCAGAGAAGCTGGAG AGGAAT-GAGCTCGAACATGCTGTGGCTTCTTGGCCTTTATGAGGATC -3′); Nluc-YAP15-R124A, anti-sense (5′-ATGAAACTGCGGCCGCTTACAACT-GCAGAGAAGCTGGAG AGGAATGAGCTGCAACATGCTGTG-GGCCTCCAGCTCCTCCTCCATC -3′); Nluc-YAP15-S127A, anti-sense (5′-ATGAAACTGCGGCCGCTTACAACTGCAGAGAAGCTGGAG AGGCATGAGCTCGAACATGCTGTGG -3′); Nluc-YAP15, sense (5′-AGCTTT GTTTAAACGCCGCCACCATGGAAG ACGCCA AAAACATAAAG-3′), anti-sense (5′-AGCTTTGTTTAAACTTACAACTGCAG AGAAGCTG-3′); 14-3-3-Cluc, anti-sense (5′-TCTACGCGTTTACACGGCGATCTTTCC-3′).

**Fusion protein production and GST pull-down assay.** pGEX-4T1 constructs were transformed into BL21 cells in 2 × YTA medium. Fusion protein expression was induced with 0.4 mM IPTG (isopropyl β-D-1-thiogalactopyranoside) for 16 h at 20 °C. Bacterial cells then were lysed by sonication. Triton X-100 (20%) was added to a final concentration of 1% and samples were gently mixed on a rotator at 4 °C for 30 min–2 h. GST-fusion proteins were purified with glutathione–sepharose beads. For GST pull-down assays, 100 µg of cell lysate from HEK293 transiently expressing 14-3-3-FLAG was added to 5 µg of GST-fusion proteins and binding was assessed by GST pulldown with protein S-agarose beads and western blotting.

**In vitro kinase assay.** Five micrograms of GST fusion protein was incubated for 20 min at 30 °C with 100 ng of recombinant active LATS2 kinase (SignalChem#L2-11G) in kinase buffer (20 mM Tris-HCl pH 7.4, 10 mM MgCl₂, 1 mM dithiothreitol, 1 mM EDTA, 100 µM sodium vanadate, 5 µM ATP). The proteins were then resolved on a 10% SDS–PAGE gel, transferred to a nitrocellulose membrane and subjected to western blotting.

**BOEC isolations.** BOEC isolations were performed according to published methods[50] with modifications in Dr. James' lab at Queen's University. Mononuclear cells were isolated in BD Vacutainer Cell Preparation Tubes (BD Biosciences), washed, and resuspended in EBM-2 (Lonza) supplemented with 10% fetal bovine serum (FBS), 1% penicillin/ streptomycin (P/S), and the EGM-2 bullet kit (complete endothelial cell growth medium (cEGM)-2). Cells ($4 \times 10^7$) per well were seeded in collagen-coated six-well plates. After 9–21 days, BOEC appeared with characteristic cobblestone morphology. BOEC with passage numbers between 3 and 9 were used for all experiments.

**Fig. 4** VEGFR is an upstream regulator of Hippo signaling. VEGFR inhibition activates LATS-BS **a** and suppresses YAP/TAZ transcriptional co-activation in HEK293A **b**. Cells were treated with each inhibitor for 4 h at 10 µM ($n = 3$). **c** VEGFR inhibition diminishes expression of YAP/TAZ targets, *CYR61/CTGF*. HEK293A were treated with inhibitors for 4 h at 10 µM. *CYR61* or *CTGF* mRNA expression was determined by qRT-PCR ($n = 3$). **d** VEGFR reduces YAP-S127 phosphorylation in HEK293 (western blotting). **e** VEGF stimulation inhibits LATS-BS activity in MCF10A stably overexpressing VEGFR1/2. MCF10A-VEGFR1/2 were transfected with LATS-BS. Cells were treated with VEGF (100 ng ml⁻¹) for the indicated times. For some samples, cells were pre-treated with axitinib at 10 µM for 3 h before VEGF treatment ($n = 3$). **f, g** VEGF increases YAP/TAZ transcriptional co-activation of *CYR61* in MCF10A-VEGFR1/2 (**f**), as well as in BOEC and MDA-MB231 (**g**). Cells were treated with 100 ng ml⁻¹ VEGF for the indicated times. *CYR61* expression was measured by qRT-PCR. For some samples, cells were pre-treated with Axitinib at 10 µM for 3 h before VEGF treatment ($n = 3$). **h-m** VEGF stimulation increases YAP/TAZ nuclear localization in MCF10A-VEGFR2 (**h, i**), MDA-MB231 (**j, k**), and BOEC (**l, m**). **h, j, l** Representative images of YAP or TAZ immunostaining are shown after treatment with 100 ng ml⁻¹ VEGF for the indicated times. Scale bar represents 15 µm. **i, k, m** YAP/TAZ subcellular localization was quantified in three separate experiments in which at least 200 cells were examined. **n, o** VEGFR2 signals through PI3K, AKT, and MEK to inhibit LATS-BS (**n**) and activate the STBS reporter (**o**). Cells were untreated or treated with VEGFR inhibitor (axitinib), PI3K inhibitor (LY294002), AKT inhibitor (triciribine), or MEK inhibitor (PD98059) at 10 µM for 4 h ($n = 3$). *$p < 0.05$ in two-sample unpaired *t*-test. **p** VEGF stimulates phosphorylation of VEGFR, AKT, and ERK in HUVEC cells, and reduces MST1 and LATS1 phosphorylation (western blotting). HUVEC cells were treated with 100 ng ml⁻¹ VEGF for the indicated times. VEGFR was inhibited using 10 µM axitinib for 3 h before VEGF treatment. All data are represented as mean ± SD

**Cell culture**. A549 (human lung adenocarcinoma), HEK293 (human embryonic kidney cells), HEK293A, HEK293T were cultured in Dulbecco's modified Eagle's medium (DMEM; Sigma D6429) containing 10% FBS, and 1% P/S (Invitrogen). For MDA-MB231 (invasive ductal breast carcinoma), DMEM was supplemented with 10% FBS, 1% P/S, and 1% non-essential amino acids (Sigma M7145). MCF10A (non-tumorigenic mammary epithelial cell line) were maintained in

DMEM/F12 medium (Sigma D6421) containing 5% Horse Serum (Sigma H1270), 0.25 mM L-glutamine (Sigma G7513), 10 μg ml$^{-1}$ Insulin (Sigma I6634), 20 ng ml$^{-1}$ hEGF (Sigma E4269), 100 ng mL$^{-1}$ Cholera toxin (Sigma C8052), 0.5 μg ml$^{-1}$ Hydrocortisone (Sigma H4001), and 1% P/S. All cells were kept at 37 °C with 5% $CO_2$. BOEC, Telo-HEC, and HUVEC (Lonza CC-2519 pooled donor) were cultured in cEGM-2 media (Lonza). *LATS1/2* and *MST1/2* knockout

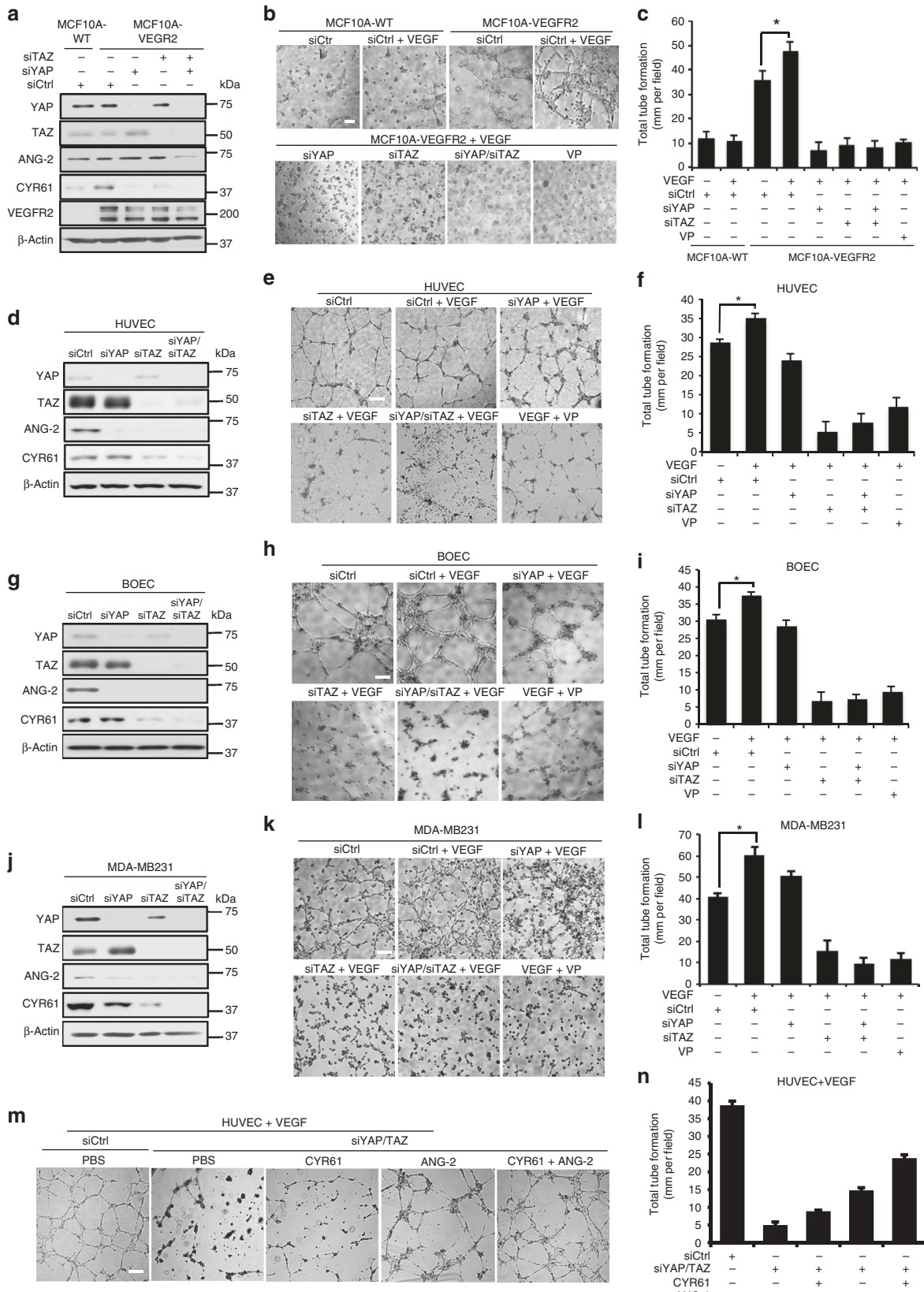

HEK293A cells were established as described[30] and cultured in DMEM/10% FBS/1% P/S. For some experiments, cells were stimulated with 100 ng ml$^{-1}$ VEGF-A (Sino Biological Inc.) after being serum-starved for 16 h.

**Lentivirus production, purification, and infection.** To produce lentiviruses, $1.5 \times 10^7$ HEK293T (passage lower than 10) cells were plated on a 150 mm plate coated with 0.1 mg ml$^{-1}$ poly-L-lysine hydrobromide (Sigma 7890). The next day, cells were transfected with 8 μg of 14-3-3-Cluc-WPI, or Nluc-YAP15-WPI or VEGFR1/2-WPI alongside 6 μg of psPAX2 (packaging vector) and 2 μg of pMD2.G (envelope vector) using Polyjet transfection reagent (Signagen SL100688) according to the manufacturer's instructions. Twenty-four hours after transfection, media was replaced with DMEM/10% FBS containing 10 mM NaButyrate. Forty-eight hours after transfection, the virus-containing media was collected, passed through a 0.45 μm filter and concentrated using Lenti-X Concentrator (Clontech 631231). To titrate the virus, $3.5 \times 10^4$ HEK293 cells were plated into a 24-well plate and were infected with a series of lentivirus dilutions (with 8 μg ml$^{-1}$ polybrene) the next day. GFP-positive cells were counted under a fluorescent microscope and the percentage of GFP-positive cells was calculated. To stably overexpress VEGFR1/2, LATS-BS, or LATS2-FLAG in cells, MCF10A, HEK293, A549, or MDA-MB231 cells were infected with lentivirus expressing cDNAs of these genes. An appropriate number of cells were plated in a six-well plate so that cells were ~ 40–50% confluent at the time of infection. Lentivirus was added with 8 μg ml$^{-1}$ polybrene at a multiplicity of infection of 0.3. Two days after infection, cells were selected with 2 μg ml$^{-1}$ puromycin or 300 μg ml$^{-1}$ hygromycin B.

**siRNA-mediated gene knockdown.** siRNA duplexes targeting *YAP* and *TAZ* were purchased from IDT. An siRNA with scrambled sequence (siCtrl) was used as a negative control. All cells were transfected with 50 nM of siRNAs using RNAi-Max (Invitrogen) according to the manufacturer's instructions. Forty-eight hours after transfection, cells were collected for in vitro angiogenesis assay or were used for protein extraction and western blot analysis.

**In vitro luciferase assay/live cells luciferase imaging.** Cells were plated in triplicate 24 h before transfection. Cells were transfected with LATS-BS, its mutants, or STBS luciferase reporter using PolyJet transfection reagent. As an internal transfection control, pRL-TK *Renilla* luciferase control reporter vector was co-transfected alongside the biosensor into each sample. Forty-eight hours after transfection, cells were collected and luciferase assay was performed using a Luciferase Assay Kit (Promega) and Turner Biosystems 20/20 luminometer. Luciferase fold change was calculated based on the ratio of luciferase activity of LATS-BS or its mutants or STBS-Luc reporter to that of vector control. The data presented is the mean of three independent experiments. For live-cell imaging, LATS-BS or a pGL3-control vector were transfected into HEK293, MDA-MB231, or A549 cells. After 48 h, cells were trypsinized and collected in a black, clear-bottomed, 96-well plate. D-luciferin (150 μg ml$^{-1}$, D-Luciferin, Potassium Salt, GoldBio UCK-250) in media was added to each well 5–10 min before imaging. Exposure time for images was ~ 3 min per plate. Imaging was performed using a LightTools Research system (Encinitas, CA) dark box and a Hamamatsu ORCA-Flash4.0 V2 digital CMOS camera over the course of 20 min, to establish optimal peak luciferase activity. The bioluminescence of the regions of interest was analyzed for total emission flux using Image Pro Plus software. The following compound treatments were used for in vitro luciferase assay and live-cell imaging: RAF inhibitor (GW5054, Cayman Chemical), ATR inhibitor (CGK733, Cayman Chemical), PI3K inhibitor 1 (GDC0941, Cayman Chemical), PI3K inhibitor 2 (LY294002, Cayman Chemical), PDK inhibitor (GSK2334470, Cayman Chemical) —10 μM for 4 h, EGF-100 ng ml$^{-1}$ for 1 h; insulin (Sigma 91077 C)—10 μg ml$^{-1}$

for 1 h; F/IBMX (Forskolin, Cayman Chemical /IBMX, Cayman Chemical)— 0.1–10 μM for Forskolin and 100 μM for IBMX for 1 h; L-α–LPA (Sigma L7260) —0.1–10 μM for 1 h, S1P-1 μM for 1 h; TPA (Cell signaling #41745)—5 nM for 1 h; and 2-deoxy glucose (Sigma #D8375)—25 mM for 1 h. For the LV200 imaging, 3.5 mM D-luciferin was added to the media culturing HEK293, A549, or MDA-MB231 cells stably expressing LATS-BS at 5–10 min before imaging. Images were captured using Olympus LV200 Bioluminescence Imager with exposure times ranging from 30 s (HEK293A) to 10 min (MDA-MB231, A549). HEK293A were purchased from Cell Biolab. All other cell lines were purchased from ATCC.

**In vivo luciferase imaging.** For cell injection studies, 12-week-old female BALB/c mice were anesthetized by exposure to 1–3% isofluorane. HEK293 cells ($3 \times 10^6$) transfected with LATS-BS alone (LATS −) or together with LATS (LATS +) were suspended in 100 μl of sterile phosphate-buffered saline (PBS) and injected into the mammary fat pad. Two days after the injection, postsurgery mice received 150 mg kg$^{-1}$ of D-luciferin (Cedarlane) dissolved in PBS by intraperitoneal injection. Imaging of ventral view was performed using a LightTools Research system (Encinitas, CA) dark box and a Hamamatsu ORCA-Flash4.0 V2 digital CMOS camera over a course of 30 min, to establish optimal peak luciferase activity. Pseudo-colored parametric overlays of BLI with anatomical reference images were dynamically constructed for each individual animal at comparative time points. The bioluminescence (BLI) of the regions of interest was then analyzed for total emission flux using Image Pro Plus software. All procedures were approved by the Queen's University Animal Care Committee in accordance with Canadian Council on Animal Care guidelines.

**Kinase inhibitor screen.** LATS-BS was transfected into HEK293A. Cells were passed into a 384-well plate the following day. Forty-eight hours after transfection, cells were treated with the Tocriscreen Kinase Inhibitor Toolbox (Tocris Bioscience 3514) with each drug administered at 10 μM in dimethyl sulfoxide (DMSO) for 4 h in duplicate. Biosensor activity was then measured by luciferase assay. Fold-change ratios were generated by comparing biosensor activity for each drug with that of DMSO-treated controls.

**RNA extraction and qRT-PCR.** Total RNA was extracted using the RNAzol RT reagent (Molecular Research Center, Inc., Cincinnati, OH, USA) according to the manufacturer's protocol. After extraction, RNA quality was evaluated by electrophoresis on 1.5% agarose gel. SuperScript III Platinum SYBR Green One-Step qRT-PCR Kit (Invitrogen) and 50 ng total RNA from cells was used for quantitative reverse transcriptase PCR (qRT-PCR). 18 S rRNA was used as internal control. The following primers were used: *rRNA*, sense (5′-TCCCCATGAACGAGGAATTCC-3′), anti-sense (5′-AACCATCCAATCGGTAGTAGC-3′); *CTGF*, sense (5′-CCCTCGCGGCTTACCGACTGG-3′), anti-sense (5′-CACAGGTCTTGGAA-CAGGCGC-3′); *CYR61*, sense (5′-AATGGAGCCTCGCATCCTATA-3′), anti-sense (5′-TTCTTTCACAAGGCGGCA-3′).

**Protein extraction, antibodies, and western blotting.** Protein were extracted using RIPA buffer. Approximately 10–20 μg of protein lysate was used to perform WB. Primary antibodies used were as follows: mouse monoclonal anti-TAZ (560235, BD Biosciences, 1 : 1,000); rabbit polyclonal anti-YAP (sc-15407, Santa Cruz H125, 1 : 1,000); mouse monoclonal anti-β-actin (A5441, Sigma, 1 : 10,000); rabbit monoclonal anti-cleaved PARP (ab32064, Abcam, 1 : 1,000); rabbit polyclonal anti-CYR61 (sc-13100, Santa Cruz, 1 : 1,000); rabbit monoclonal anti-luciferase (ab185923, Abcam, 1 : 1,000); rabbit polyclonal anti-HA (H6908, Sigma, 1 : 1,000); mouse monoclonal anti-FLAG (F1804, Sigma, 1 : 1,000); mouse monoclonal anti-Myc (9E10) (11667203001,

**Fig. 5** VEGFR regulates angiogenesis and tumor VM through YAP/TAZ in vitro. **a** Transient knockdown of YAP and/or TAZ in MCF10A overexpressing VEGFR2 decreases expression of ANG-2 and CYR61. Western blotting exposures indicate relative expression of YAP and TAZ. **b**, **c** VEGFR2 overexpression and VEGF treatment increases tube formation by MCF10A through YAP/TAZ. YAP and/or TAZ were transiently knocked down by siRNA in MCF10A stably overexpressing VEGFR2 and subjected to tube-formation assay on Matrigel 48 h after transfection alongside wild type MCF10A. For some conditions, cells were stimulated with 100 ng ml$^{-1}$ VEGF or were treated with 100 nM verteporfin for the duration of the tube formation assay. Representative images are shown in **b**. Scale bar denotes 200 μm. Total tube formation was quantified in **c** (n = 3). *p < 0.05 in two-sample unpaired t-test. **d–l** YAP/TAZ are critical for VEGF-induced angiogenesis in HUVEC (**d–f**) and BOEC (**g–i**), as well as for vasculogenic mimicry in MDA-MB231 (**j–l**). **d**, **g**, **j** YAP and/or TAZ were transiently knocked down by siRNA in each cell line which reduced ANG-2 and CYR61 expression. Western blotting exposures indicate relative expression of YAP and TAZ for each cell line. **e**, **h**, **k** Representative images of tube formation on Matrigel assessed 48 h after transfection. For some conditions, cells were stimulated with 100 ng ml$^{-1}$ VEGF or were treated with 100 nM verteporfin for the duration of the tube formation assay. Scale bar denotes 200 μm. Total tube formation was quantified in **f**, **i**, **l** (n = 3). *p < 0.05 in two-sample unpaired t-test. **m**, **n** Exogenous CYR61 and ANG-2 can partially rescue tube formation in YAP/TAZ knockdown HUVEC. YAP and/or TAZ were transiently knocked down by siRNA in HUVEC. Tube formation on Matrigel was assessed 48 h after transfection. For some conditions, cells were stimulated with 100 ng ml$^{-1}$ VEGF, 200 ng ml$^{-1}$ CYR61, and/or 200 ng ml$^{-1}$ANG-2 for the duration of the tube formation assay. Scale bar denotes 200 μm. Representative images are shown in **m** and total tube formation was quantified in **n**. (n = 3). All data are represented as mean ± SD

Roche, 1 : 1,000); rabbit polyclonal anti-phospho YAP (S127) (4911, Cell Signaling, 1 : 1,000), rabbit monoclonal anti-LATS1 (3447, Cell Signaling, 1 : 1,000), rabbit monoclonal anti-ERK1/2 (4695, Cell Signaling, 1 : 1,000), mouse monoclonal anti-AKT (2967, Cell Signaling, 1 : 1,000), rabbit monoclonal anti-phospho-AKT (Ser473) (4060, Cell Signaling, 1 : 1,000), rabbit monoclonal anti-phospho-ERK (4370, Cell Signaling, 1 : 1,000), rabbit monoclonal anti-phospho-MST1 (Ser183) (3681, Cell

Signaling, 1 : 1,000), rabbit monoclonal anti-MST1 (3682, Cell Signaling, 1 : 1,000), rabbit monoclonal anti-phospho-LATS1 (Thr1079) (8654, Cell Signaling, 1 : 1,000), rabbit monoclonal anti-phospho-VEGFR2 (3770, Cell Signaling, 1 : 1,000), rabbit monoclonal anti-VEGFR2 (2479, Cell Signaling, 1 : 1,000), mouse monoclonal anti-phospho-tyrosine 4G10 (no catalog number, Millipore, 1 : 1,000), and polyclonal goat

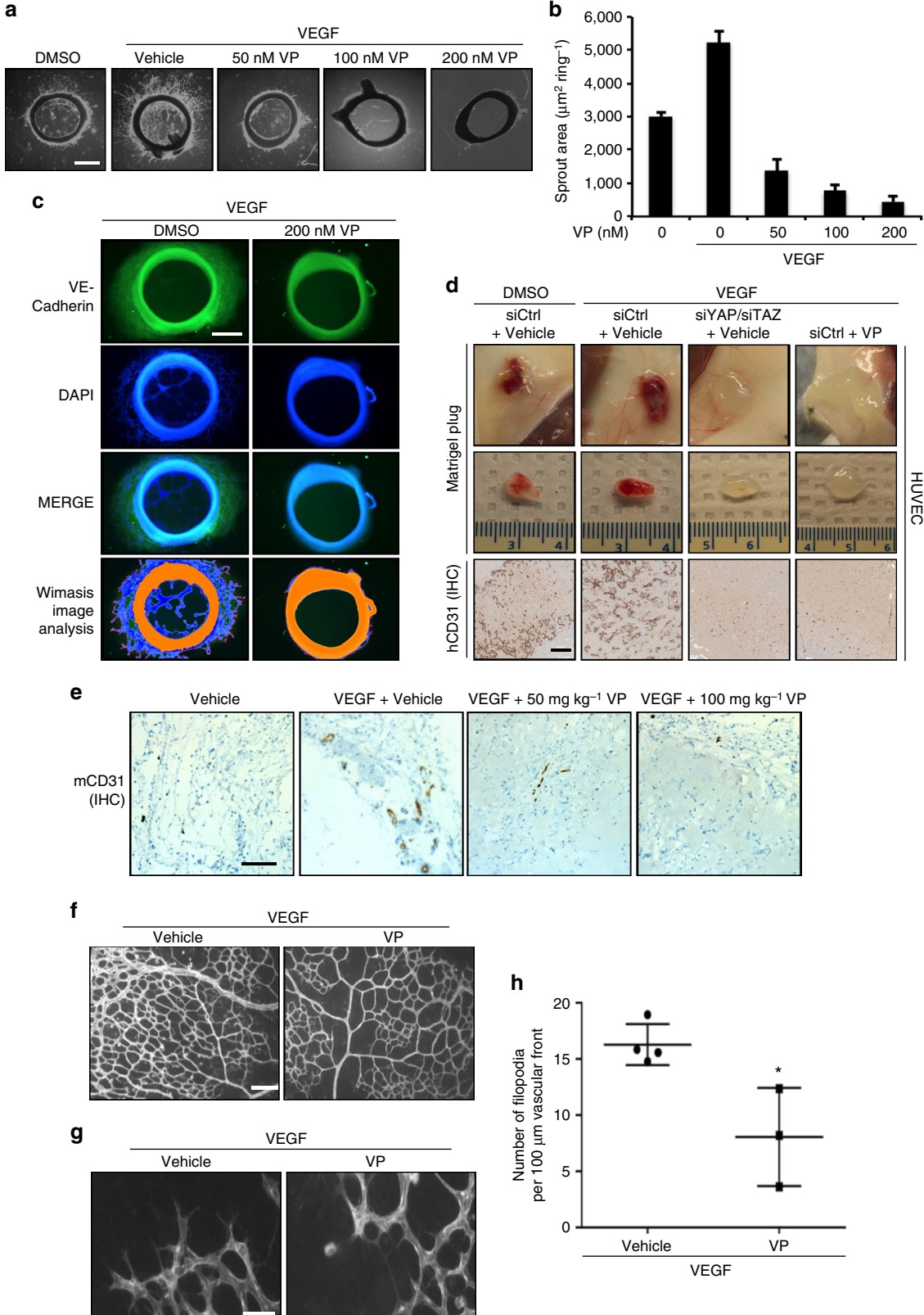

anti-ANG2 (AF623, R&D, 1 : 1,000). Uncropped images of western blottings are available in Supplementary Fig. 6.

**Immunofluorescence**. Cells were seeded in a 24-well plate with poly-L-lysine and fibronectin-coated coverslips in an appropriate number to achieve 40–50% confluency at the time of immunostaining. Cells were serum-starved for 16 h and untreated or treated with 100 ng ml$^{-1}$ VEGF-A for 15–60 min. Cells were fixed in 5% formaldehyde for 15 min and were permeabilized in 0.2% Triton X-100 in PBS for 2 min. Cells were subsequently incubated with primary and secondary antibodies diluted in PBS with 1% bovine serum albumin (BSA) and 10% goat serum. The same conditions were used for fixation, permeabilization, and staining of aorta ring. Antibodies used were: rabbit polyclonal anti-YAP (Santa Cruz H125, 1 : 100), mouse monoclonal anti-TAZ (BD Biosciences, 1 : 400), mouse monoclonal anti-VE-cadherin (Santa Cruz F-8, 1 : 100), Alexa Fluor (AF)-555 rabbit anti-mouse IgG and AF-488 goat anti-rabbit IgG. Images were captured with a Nikon TE-2000U (Melville, NY, USA) inverted microscope. Subcellular localization of YAP or TAZ was quantified from three separate experiments in which at least 200 cells were examined from the immunostaining.

**Immunohistochemistry**. Matrigel plugs were embedded in paraffin before cutting into 5 μm sections. Immunohistochemistry (IHC) signals were developed using rabbit monoclonal antibodies against human CD31 (1 : 100, Cell Signaling) or mouse CD31 (1 : 100, Cell Signaling). IHC was performed on a Ventana auto-stainer Discovery XT platform, using Ventana deparafinization and antigen retrieval protocols. EDTA antigen retrieval, pH 8, was used.

**Matrigel tube formation assay**. Four hours before angiogenesis assay, cell media was replaced with media containing 1% serum. μ-Slide Angiogenesis plates (Ibidi, 81506) were coated with 10 μl Matrigel and were allowed to polymerize for at least 30 min before use. Cells ($1 \times 10^4$) ($2 \times 10^4$ for MDA-MB231) were seeded into each well in triplicate. After 8 and 20 h at 37 °C, cells were imaged at ×4 magnification on a Nikon TE-2000U inverted microscope. In some experiments, 100 ng ml$^{-1}$ VEGF-A or freshly prepared VP (Sigma, SML0534) were added into the medium. VP-treated samples were protected from light for the duration of the experiment. Tube formation was quantified using ImageJ[50,51] and online WimTube software (Wimasis Image Analysis, Ibidi)[52].

**Aortic ring angiogenesis assay**. The thoracic aorta was dissected from Sprague–Dawley rats, cleaned in PBS + 1% penicillin/streptomycin cocktail and cut

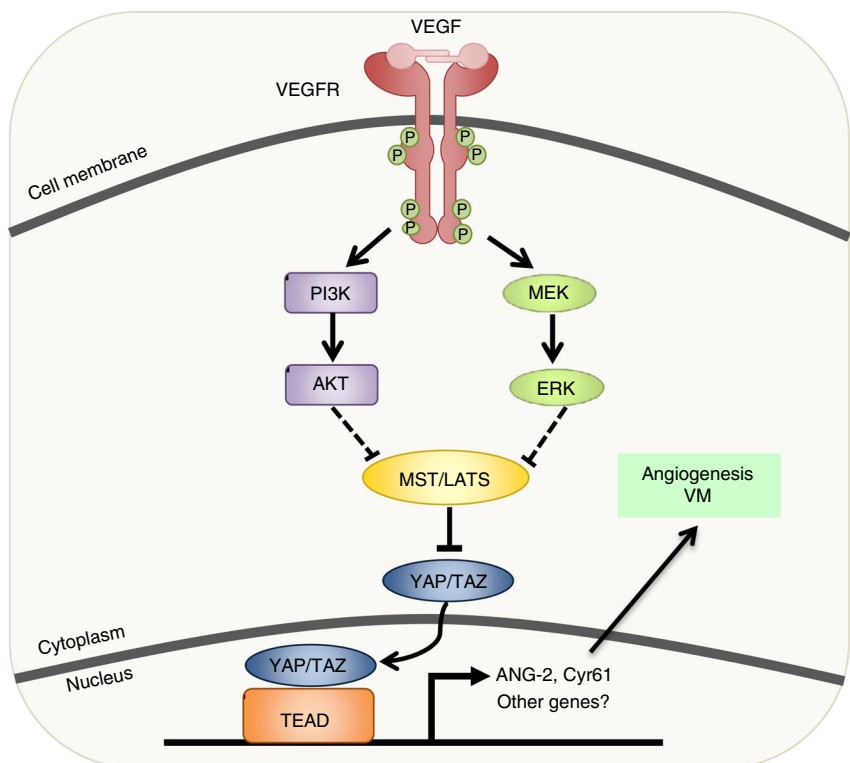

**Fig. 7** Model for VEGFR and Hippo signaling in angiogenesis/VM. When VEGF binds to its receptor, VEGFR, signaling through PI3K and MAPK is initiated. This leads to the inhibition of MST/LATS and subsequent activation of YAP/TAZ. YAP and TAZ induce ANG-2 and CYR61 expression, leading to enhanced angiogenesis and vasculogenic mimicry in endothelial and tumor cell lines, respectively

**Fig. 6** YAP/TAZ are mediators of VEGF-induced angiogenesis ex vivo and in vivo. **a–c** Pharmacological inhibition of YAP/TAZ reduces angiogenesis ex vivo in a rat aorta model. Sections of aorta were cultured for 7 days in Matrigel with 100 ng ml$^{-1}$ VEGF and the indicated concentrations of VP. Representative images are shown in **a**. Sprout area is quantified in **b**. Scale bar denotes 500 μm. **c** Immunostaining of aorta sections demonstrates that outgrowths are positive for VE-cadherin. Scale bar denotes 300 μm. Wimasis image analysis software was used to visualize sprouts (n = 3). **d** Transient knockdown of YAP/TAZ or pharmacological inhibition of YAP/TAZ with VP reduces angiogenesis by HUVECs in vivo in Matrigel plug experiments. Five million cells were injected subcutaneously into mice with Matrigel and 200 ng mL$^{-1}$ VEGF. VP was administered by intraperitoneal injection every other day. Plugs were excised after 1 week. Representative images are shown in the top two rows. In the bottom panels, angiogenesis in the Matrigel plugs was stained by IHC for the human endothelial cell marker hCD31. Scale bar denotes 500 μm. **e** YAP/TAZ inhibition reduces endogenous angiogenesis in vivo in Matrigel plug experiments. Matrigel plugs with 200 ng ml$^{-1}$ VEGF were implanted subcutaneously in mice. VP was administered by intraperitoneal injection every other day. Plugs were excised after 2 weeks. Angiogenesis was assessed by IHC staining for mouse mCD31 endothelial cell marker. Scale bar denotes 500 μm. **f–h** YAP/TAZ inhibition diminishes angiogenesis in vivo in a mouse retinal model. Mice were injected with 1 mg kg$^{-1}$ VEGF with or without 100 mg kg$^{-1}$ VP at postnatal day 3 (P3) and 4 (P4). At P5, retinal blood vasculature was stained. Representative images of the retinal vessel density are shown in **f**, whereas **g** shows images of the vascular front. Scale bar denotes 100 μm **f** or 30 μm **g**. Number of filopodia (active angiogenesis) is quantified in **h**. Each data point represents the average of two retinas from a single mouse (n = 4 for control, n = 3 for VP). *p < 0.05 in two-sample unpaired t-test. All data are represented as mean ± SD

into 0.5 mm ring segments, washed with PBS plus 1% penicillin/streptomycin cocktail, and embedded in growth factor-reduced Matrigel in a 24-well cell culture plate. Embedded rings were covered in EBM-2 media containing, 1% FBS, 1% P/S cocktail, and incubated at 37 °C and 5% $CO_2$ for 24 h before treatment with vehicle or VP. VP-treated samples were protected from light throughout the assay. Photographs were taken every 24 h using a phase-contrast microscope (Olympus CKX41) for up to 7 days. Endothelial cell sprouting was quantified using Image-Pro[50] and online Wimsprout software (Wimasis Image Analysis, Ibidi)[53]. All procedures were approved by the Queen's University Animal Care Committee in accordance with Canadian Council on Animal Care guidelines[54].

**In vivo Matrigel plug assay**. In vivo analysis of angiogenesis was evaluated using a Matrigel plug assay. Two different Matrigel plug experiments were performed. First, $5 \times 10^6$ HUVEC-siCtrl cells ± VEGF or HUVEC-siYAP/siTAZ cells + VEGF were mixed in 250 μl of Matrigel and injected subcutaneously into the flanks of 12-week-old Rag2$^{-/-}$; Il2rg$^{-/-}$ mice. One-day post Matrigel injections, mice were divided into groups and treated with vehicle or 100 mg kg$^{-1}$ VP. Treatments were administered via intraperitoneal injections every other day. Three mice were used for each treatment condition. On day 7, mice were euthanized and Matrigel plugs containing HUVEC cells were collected. In the second experiment, a total of 200 μl of cold growth factor reduced phenol-red free Matrigel ± VEGF (200 ng ml$^{-1}$) was injected subcutaneously. One-day post Matrigel injections, mice were divided into three treatment groups: control (DMSO), 50 mg kg$^{-1}$ VP dose, and 100 mg kg$^{-1}$ high VP. Treatments were administered via intraperitoneal injections every other day. Three mice were used for each treatment condition. On day 14, plugs were collected. All Matrigel plugs were exposed and digitally photographed followed by excision. Plugs were then fixed in 4% paraformaldehyde and paraffin-embedded for sectioning. All procedures were approved by the Queen's University Animal Care Committee in accordance with Canadian Council on Animal Care guidelines.

**Retina whole-mount immunofluorescence**. Mice were treated at postnatal day 3 and 4 with 1 mg kg$^{-1}$ VEGF and/or 100 mg kg$^{-1}$ VP by intraperitoneal injection. On postnatal day 5, pups were euthanized and eyes were collected and fixed in 4% paraformaldehyde (15710-S, Electron Microscopy Sciences, Pennsylvania, USA) for 2 h at room temperature. Retinas were isolated by dissecting away the cornea, sclera, lens, vitreous body, and hyaloid vessels. Retinal vessels were stained with Alexa Flour 594-conjugated *Griffonia simplicifolia* Isolectin B4 (IB4; I21413, ThermoFisher Scientific, Maine, USA) after 2 h of blocking in PBS (PBS404, Bio-Shop, Burlington, Canada) with 0.2% BSA (ALB006.5, BioShop) and 0.5% Triton-X (TRX506, BioShop), and washing in PBS with 1% Triton-X. Photomicrographs of the retinal vasculature were taken using an Imager Z1m microscope and Axiovision MR3 camera (Zeiss, Toronto, Canada) at ×100 and ×200 magnification. ImageJ and Angiotool[55] were used for analysis. Two retinas from a single pup were treated as technical replicates and averaged. All procedures were approved by the Queen's University Animal Care Committee in accordance with Canadian Council on Animal Care guidelines.

**Statistical analysis**. For all graphs, data are presented as mean ± SD. Statistical comparisons between groups were performed using a Student's *t*-test or analysis of variance.

**Data availability**. The authors declare that all data supporting the conclusions of this study are presented within the paper and the supplementary information files and are available from the authors.

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

## Acknowledgements

This work was supported by grants from Canadian Institute of Health Research (CIHR#119325, 148629), Canadian Breast Cancer Foundation (CBCF) to X.Y. This work was also supported by the Roswell Park Cancer Institute and National Cancer Institute (NCI) grant number P30 CA016056, Roswell Park Alliance Foundation, and in part by the National Cancer Institute (NCI) R01 CA207504, the American Cancer Society Research Scholar Grant RSG-14-214-01-TBE to J.Z. T.A. is supported by the Vanier Canada Graduate Scholarship and Ontario International Graduate Scholarship. H.J.J.v.R is supported by a Queen Elizabeth II Graduate Scholarship in Science and Technology. We thank Drs David Lillicrap and Paula James for providing the HUVEC, Telo-HEC, and BOEC cells; Alison Michels and Soundarya Selvam for helping in HUVEC and BOEC cell culture and angiogenesis analysis; Mina Ghahremani for help in making some figures; Drs Peter Greer, Kazem Nouri, and Amin Tashakor for reading the manuscript; Dr Javad Hashemi from Queen's School of Computing for helping with data analysis; Lee Boudreau for performing IHC; Jia Yue Amelia Shi for help with mouse work; as well as Roswell Park Cancer Institute Core Facility for performing the kinase inhibitor screen.

## Author contributions

T.A. and H.J.J.v.R. performed most of the experiments on molecular cloning and mutagenesis, siRNA knockdown, luciferase assays, establishing VEGFR- and LATS-BS-overexpressing stable lines, VEGF treatments and mRNA extractions, western blotting, immunostaining, and tube formation assays. T.A., E.D.L. and C.J.B.N. designed and performed BLI analysis in cells and mice, as well as in vivo Matrigel plug assay. F.P., B.N. and A.C. designed and performed LV200 imaging experiments and data analysis. Y.H. performed real-time qRT-PCR. H.S. and J.Z. performed kinase inhibitor screening experiments. B.Y. established LATS2-expressing cell line. T.A. and A.G. designed and performed aortic ring angiogenesis assay. V.R.K., A.C. and E.D.L. designed and performed retina whole-mount immunofluorescence. K.L.G. provided the *LATS1/2* and *MST1/2* knockout HEK293A cells. T.A., B.A.C. and X.Y. designed experiments. T.A., H.J.J.v.R. and X.Y. wrote the manuscript.

## Additional information

**Competing interests:** The authors declare no competing interests.

