## [Peer Review File · Nature Communications]

Reviewers' comments:

Reviewer #1 (Remarks to the Author):

This manuscript describes the creation of a sensor of LATS activity, based on the split-luciferase approach. The authors use this sensor for a small-scale screen of kinase inhibitors, which leads them to identify VEGF Receptors as regulators of LATS activity. They confirm that VEGF can influence Hippo signaling, and show that YAP & TAZ contribute to tubulogenesis induced by VEGF. Altogether it's an interesting study that contributes to the Hippo field by presenting a new tool for investigating Hippo signaling in vivo, by identifying another upstream regulator of the pathway, and by implicating Hippo signaling in vasculogenesis. Thus I think it makes a significant contribution. The experimental analysis is convincing and includes appropriate quantitation.

I have only a couple minor suggestions for improvement:

- The authors report that "our kinase inhibitor screen revealed that inhibitors of MEK, PI3K and AKT/PKB activated the LATS-BS (Fig. 3c)" These kinases, and pathways that regulate these kinases (eg EGFR, Insulin) have already been previously identified as modulators of Hippo signaling, and the authors should cite those earlier studies.

- Fig 2 legends, the authors state "LATS-BS can be used to observe ... active LATS kinase subcellular localization (g)." This is incorrect. The LATS-BS shows the location of an artificial substrate, which could potentially diffuse through the cell; it doesn't report the location of active LATS kinase.

Reviewer #2 (Remarks to the Author):

The authors show that VEGFR activation negatively regulates the hippo pathway in endothelial and cancer cells. Specifically, they show that inhibiting VEGFR activates a biosensor that measures the activation of the LATS kinase. In agreement with LATS kinase activation being known to inactivate the transcriptional regulators YAP and TAZ, they further show that the level of gene expression for several YAP/TAZ targets is altered. This is an interesting observation of potential interest for both the angiogenesis and cancer fields. However, the study is not at all conclusive to actually demonstrate such significance. Thus, it has not been shown that the VEGFR regulation of hippo pathway signalling is relevant for angiogenesis or cancer, because relevant in vivo experiments have not been performed. An in vitro tubulogenesis assay does not equate to angiogenesis or vasculogenic mimicry of cancer cells. As such, the title of the manuscript is misleading, and the abstract and discussion inaccurate.

The authors also present an interesting and potentially valuable research tool to measure hippo pathway activation, a biosensor based on the phosphorylation of the hippo pathway component YAP by LATS1/2. The biochemical experiments are sound, although the in vivo use of the biosensor is not clear – simply showing that one can image activation is not biologically relevant.

The conclusion of a VEGF/VEGFR-LATS-YAP/TAZ-ANG-2/Cyr61 signaling axis is an overstatement based on the data that are presented.

The relevance of their findings to human disease are grossly over-stated and poorly explained.

In summary, there is value of the study, but it is limited to the discovery of a tool useful to study hippo signalling and the description of signal transduction cascade target for VEGF signalling, but the impact of the study is limited, and its significance for important biological pathways not at all clear.

Specific comments:

The authors over-use the term 'novel'.

Abstract – what do the authors mean with “we recently developed a ...biosensor”? Is this the first time this is presented in a manuscript or not?

It is not clear where the introduction ends and the results start.

Page 3 – “a few regulatory factors of the hippo pathway have been uncovered’ and the state actin dynamics, cell-matrix stiffness and cell contact as examples – but these are not regulators – they are cellular processes.

Page 4 - It is not clear how the discovery of hippo signalling being regulated by VEGF signaling would help in the treatment of diseases that are currently modulated by anti-VEGF or anti-VEGFR strategies. This section is very vague.

Page 4 – what is meant with ‘confounding signals by posttranslational modifications of YAP by other regulators’?

Page 5 – “In addition, the basal LATS-BS signal was reduced by knockout of endogenous MST1/2 in HEK293A (50% reduction) or more dramatically by LATS1/2 knockout (~90% reduction)” - no mention of the knockdown method here or in the methods section. Why is there a much higher reduction in LATS-BS signal in the LATS1/2 KO compared with MST1/2 KO? Is something else able to phosphorylate, and activate, LATS1/2 when MST1/2 is not present?

Page 6 – what is meant with ‘our newly developed method for the LV200BL1 system’? What is the important information here? How is this useful to the reader?

Page 8 – the term “incredibly’ is inappropriate

Page 9 – “We used the same approach to examine the role of YAP and TAZ during angiogenesis using human umbilical vein endothelial cells (HUVEC) and BOECs. In these cells, endogenous YAP expression was barely detectable while TAZ was highly expressed (Fig. 5d, g). “ It would be important to investigate the relative YAP and TAZ expression levels in all of the cell types used in this study.

Page 9 - “Consequently, TAZ knockdown had a more substantial effect on blocking angiogenesis than YAP knockdown in both HUVECs and BOECs.” In contrast, they show that TAZ is barely detectable in MCF10A cells, whereas YAP is high. Yet, KD of each has the same effect on tube formation...why might this be?

Page 9 – “While knockdown of both YAP and TAZ reduced both ANG-2 and CYR61 in MCF10A-VEGFR2 cells, TAZ knockdown alone was sufficient for loss of ANG-2 and CYR61 in HUVEC, BOEC, and MDA-MB-231 cells (Fig. 5a, 5d, 5g, 5j).” I do not understand these statements, and who overexpression was used for some experiments.

The authors do not to explore whether VEGFR could be inhibiting a protein upstream of LATS1/2 (eg. MST1/2) – the inhibition of LATS1/2 could merely be a knock-on effect MST1/2 inhibition.

The authors state: “ Thus, ANG-2 and CYR61 may be critical downstream gene targets of YAP/TAZ mediating angiogenesis and vasculogenic mimicry (Fig. 5m). “ This is a possibility, but it is also likely that there are many other YAP/TAZ targets with altered expression levels downstream of VEGFR signalling that additionally, or alternatively, play a role in either endothelial or cancer cells.

There is no evidence provided to argue that ANG-2 and Cyr61 are the specific hippo effectors regulating angiogenesis in vitro or in vivo.

The authors could express ANG-2 or CYR61 to see if this would rescue the reduced tube formation seen in YAP/TAZ KD cells and determine whether formation is reduced if ANG-2 and/or CYR61 are knocked down. However, these in vitro assays would only raise the possibility of a role in angiogenesis.

Although the authors mention MST1/2 at the beginning of the paper, there has been no mention of this, or other proteins further upstream in the hippo pathway, after that. How can they be sure that VEGFR inhibits LATS and not MST1/2 etc, which then inhibit LATS as a knock-on effect?

Figure 4: Contrary to what is said in the figure legend, data are represented as percentages in nucleus/cytoplasm/both, not as mean. How many independent experiments were carried out (as opposed to cells analysed or duplicates etc).

The authors should mention: Phosphorylation of angiominin by Lats1/2 kinases inhibits F-actin binding, cell migration, and angiogenesis. Dai X, She P, Chi F, Feng Y, Liu H, Jin D, Zhao Y, Guo X, Jiang D, Guan KL, Zhong TP, Zhao B. *J Biol Chem*. 2013 Nov 22;288(47):34041-51. doi: 10.1074/jbc.M113.518019. Epub 2013 Oct 8. PMID: 24106267

Reviewer #3 (Remarks to the Author):

The manuscript by Azat and colleagues from Xiaolong Yang's laboratory reports on VEGFR as a possible upstream receptor that modulates Hippo-YAP pathway. The authors used a cleverly-designed LATS biosensor to monitor the effect of kinase inhibitors on the phosphorylation of YAP on Serine 127 residue that if it is decorated by phosphate serves generally as the inactivation signal for YAP protein. The work is novel and interesting; the Hippo-YAP pathway is being connected to the angiogenic propensity of tumor formation.

The following changes are suggested to improve the manuscript:

1. It would be important to monitor the biosensor fusion protein itself for changes in tyrosine phosphorylation status to eliminate a possibility that the basic constitutive phosphorylation of the biosensor is changed, for example, by inhibiting VEGFR or EGFR.
2. It seems that there is "a back box" in terms of the presented data on the actual mechanism by which inhibition of VEGFR activates LATS kinase. Therefore it is suggested to be more careful using statements like "a novel VEGFR-PI3K/MAPK-LATS-YAP/TAZ-Cyr61/CTGF signaling axis", or a new signaling pathway.
3. Serine 128 of YAP is a target of NEMO-like kinase that overrides Serine 127 phosphorylation of YAP. This could be discussed briefly, if space allows.
4. Basu, Subham and colleagues reported in 2003 (in *Mol. Cell* 11, pages 11-23) that YAP could be directly phosphorylated by AKT kinase on Serine 127. The scenario of AKT-YAP signaling and its regulation by VEGFR inhibition could be verified in cell models, or at least discussed.
5. For sure the report is exciting in terms of a novel functional connection. To make this communication lucid, I would suggest that the length of the text be shortened by 40% focusing on the most important findings and selecting additional figures from the main body of the report to be

placed in Supplementary Data. The presented here data on the LATS-YAP sensor are intriguing. The LATS-YAP sensor should be a valuable reagent for further exploration by the Hippo-YAP research community. It would be good if it is deposited to ADDGENE company for fast distribution among researchers.

Response to Reviewers' comments:

Reviewer #1:

This manuscript describes the creation of a sensor of LATS activity, based on the split-luciferase approach. The authors use this sensor for a small-scale screen of kinase inhibitors, which leads them to identify VEGF Receptors as regulators of LATS activity. They confirm that VEGF can influence Hippo signaling, and show that YAP & TAZ contribute to tubulogenesis induced by VEGF. Altogether it's an interesting study that contributes to the Hippo field by presenting a new tool for investigating Hippo signaling in vivo, by identifying another upstream regulator of the pathway, and by implicating Hippo signaling in vasculogenesis. Thus I think it makes a significant contribution. The experimental analysis is convincing and includes appropriate quantitation.

We thank the reviewer for recognizing the significant contribution our findings will make to the field.

I have only a couple minor suggestions for improvement:

- The authors report that “our kinase inhibitor screen revealed that inhibitors of MEK, PI3K and AKT/PKB activated the LATS-BS (Fig. 3c)” These kinases, and pathways that regulate these kinases (eg EGFR, Insulin) have already been previously identified as modulators of Hippo signaling, and the authors should cite those earlier studies.

We have highlighted these known regulators in the text and added the appropriate citations.

- Fig 2 legends, the authors state “LATS-BS can be used to observe ... active LATS kinase subcellular localization (g).” This is incorrect. The LATS-BS shows the location of an artificial substrate, which could potentially diffuse through the cell; it doesn't report the location of active LATS kinase.

We agree that this is a possibility and have modified the Figure 2g legend to remove this statement.

Reviewer #2:

The authors show that VEGFR activation negatively regulates the hippo pathway in endothelial and cancer cells. Specifically, they show that inhibiting VEGFR activates a

biosensor that measures the activation of the LATS kinase. In agreement with LATS kinase activation being known to inactivate the transcriptional regulators YAP and TAZ, they further show that the level of gene expression for several YAP/TAZ targets is altered. This is an interesting observation of potential interest for both the angiogenesis and cancer fields. However, the study is not at all conclusive to actually demonstrate such significance. Thus, it has not been shown that the VEGFR regulation of hippo pathway signalling is relevant for angiogenesis or cancer, because relevant *in vivo* experiment have not been performed. An *in vitro* tubulogenesis assay does not equate to angiogenesis or vasculogenic mimicry of cancer cells. As such, the title of the manuscript is misleading, and the abstract and discussion inaccurate.

We sincerely appreciate the reviewer's thoughtful critique and useful suggestions. To address the reviewer's concerns, we have performed several additional functional experiments *ex vivo* and *in vivo* and obtained new data confirming the impact of Hippo pathway signalling on angiogenesis, which strengthens our original conclusions. First, we examined the roles of YAP/TAZ in VEGF-induced *ex vivo* angiogenesis using the rat aortic ring angiogenesis model. In this experiment, pharmacological inhibition of YAP/TAZ using verteporfin (VP) significantly decreased aortic ring sprouting angiogenesis by rat endothelial cells in a dose dependent manner (Fig. 6a, b, c, Supplementary Fig. 5). Second, we further assessed *in vivo* angiogenesis using two different models: 1) we compared VEGF-induced angiogenesis in matrigel plugs containing human primary endothelial cells (HUVECs), HUVECs genetically modified to suppress YAP/TAZ (siYAP/siTAZ) or HUVECs in mice treated systemically with VP. We have added the first evidence that both transient knockdown of YAP/TAZ, as well as VP-mediated pharmacological inhibition, significantly decrease angiogenesis (Fig. 6d); 2) We used a matrigel plug model of endogenous *in vivo* angiogenesis of mouse endothelial cells and add the new data showing systemic VP treatment decreases mouse endothelial cell angiogenesis (Fig. 6e). Finally, we also evaluated angiogenesis using the well-established mouse retinal angiogenesis model. Systemic VP treatment significantly decreased VEGF-induced retinal angiogenesis in mice (Fig. 6f-h). Therefore, taken together with our previous findings, we conclude the Hippo pathway plays a critical role in VEGF-induced angiogenesis *in vivo*.

The authors also present an interesting and potentially valuable research tool to measure hippo pathway activation, a biosensor based on the phosphorylation of the hippo pathway component YAP by LATS1/2. The biochemical experiments are sound, although the *in vivo* use of the biosensor is not clear – simply showing that one can image activation is not biologically relevant.

We thank the reviewer for acknowledging the potential value of our LATS biosensor. While future studies confirming the *in vivo* utility of our biosensor (e.g. making transgenic mice expressing the biosensor) are beyond the scope of our current manuscript, we have revised our manuscript wording accordingly (e.g. potentially) to address this concern.

The conclusion of a VEGF/VEGFR-LATS-YAP/TAZ-ANG-2/Cyr61 signaling axis is an overstatement based on the data that are presented.

To address this concern, we have deleted this or similar sentence throughout our text and modified our wording to more accurately reflect our findings.

The relevance of their findings to human disease are grossly over-stated and poorly explained.

As above, we have modified our discussion of these areas to accurately reflect and explain the clinical relevance of our work.

In summary, there is value of the study, but it is limited to the discovery of a tool useful to study hippo signalling and the description of signal transduction cascade target for VEGF signalling, but the impact of the study is limited, and its significance for important biological pathways not at all clear.

The Hippo and VEGF/VEGR pathways are important signalling cascades that play central roles in many important biological processes. In this study, we unveil a novel link between VEGF and the Hippo pathways in angiogenesis. Since angiogenesis plays important roles in a wide range of human physiology and diseases, our discovery is significant in exploring the roles of VEGF/VEGF-Hippo signalling axis in these processes. In addition, this VEGF-Hippo link may be also important in many other biological processes regulated by VEGFR and Hippo. Moreover, we believe that we have presented a new tool for investigating Hippo signaling. Given the Hippo pathway is emerging as a central player in development (e.g. organ size control, 3D body shape, and early embryo development), cancer (tumorigenesis, metastasis, drug resistance, and immune evasion), regenerative medicine (stem cell renewal and differentiation and tissue homeostasis), heart development and disease (cardiomyocyte proliferation and heart infarction/cardiac injury) and neuronal regulation (neural fate and dendrite tiling), our LATS biosensor may prove invaluable for investigating the roles of Hippo signaling in all of these important biological processes.

Specific comments:

The authors over-use the term ‘novel’.

We have amended the text to reduce the use of this term.

Abstract – what do the authors mean with “we recently developed a ...biosensor”? Is this the first time this is presented in a manuscript or not?

This is indeed the first time our biosensor has been described and we have corrected the statement to clarify this.

It is not clear where the introduction ends and the results start.

We have added headings to distinguish these sections.

Page 3 – “a few regulatory factors of the hippo pathway have been uncovered’ and the state actin dynamics, cell-matrix stiffness and cell contact as examples – but these are not regulators – they are cellular processes.

This has been corrected.

Page 4 - It is not clear how the discovery of hippo signalling being regulated by VEGF signaling would help in the treatment of diseases that are currently modulated by anti-VEGF or anti-VEGFR strategies. This section is very vague.

We have deleted the statement in the text so that it will not be over-stated.

Page 4 – what is meant with ‘confounding signals by posttranslational modifications of YAP by other regulators’?

We have revised our statement to clarify that our biosensor was designed to avoid confounding posttranslational modifications of YAP (i.e. phosphorylation at a different site, ubiquitination etc.). These modifications are avoided in our biosensor because they may otherwise affect YAP stability or function independent of LATS, which could potentially affect our LATS-BS stability and, more importantly, specificity for measuring LATS activity.

Page 5 – “In addition, the basal LATS-BS signal was reduced by knockout of endogenous MST1/2 in HEK293A (50% reduction) or more dramatically by LATS1/2 knockout (~90% reduction)” - no mention of the knockdown method here or in the methods section. Why is there a much higher reduction in LATS-BS signal in the LATS1/2 KO compared with MST1/2 KO? Is something else able to phosphorylate, and activate, LATS1/2 when MST1/2 is not present?

We have previously published the use of these CRISPR knockout cell lines (Meng *et al.*, *Nat Commun*, 2015). Indeed, other factors do activate LATS1/2 independent of MST1/2 (Meng *et al.*, 2015); for example, when we knock out MST1/2 only one major upstream regulator of LATS1/2 activation is inhibited, which explains why we see a more dramatic reduction in LATS-BS activity with LATS1/2 knockout compared to MST1/2 knockout (Fig. 1g).

Page 6 – what is meant with ‘our newly developed method for the LV200BL1 system’? What is the important information here? How is this useful to the reader?

Inclusion of this information is intended to help the readers understand that the LATS-BS bioluminescence is detectable under the microscope. However, this requires the use of a specialized LV200 system. We have modified our wording to clarify this point.

Page 8 – the term “incredibly” is inappropriate

We have changed “incredibly” to “notably”.

Page 9 – “We used the same approach to examine the role of YAP and TAZ during angiogenesis using human umbilical vein endothelial cells (HUVEC) and BOECs. In these cells, endogenous YAP expression was barely detectable while TAZ was highly expressed (Fig. 5d, g). “ It would be important to investigate the relative YAP and TAZ expression levels in all of the cell types used in this study.

We have ensured that the Western blot exposures we used for our figures reflect the relative expression levels of YAP and TAZ and this is already stated in our text. As shown in Figure 5a,d,g,j, MCF10As have high YAP and low TAZ; HUVEC/BOECs have low YAP and high TAZ; and MDA-MB-231s have high YAP and TAZ.

Page 9 - “Consequently, TAZ knockdown had a more substantial effect on blocking angiogenesis than YAP knockdown in both HUVECs and BOECs.” In contrast, they show that TAZ is barely detectable in MCF10A cells, whereas YAP is high. Yet, KD of each has the same effect on tube formation...why might this be?

To address the reviewer’s question, these data are our observations, which we interpret to mean that YAP/TAZ double knockdown inhibits tube formation. We currently do not know why knockdown of TAZ inhibits tube formation in all of the cell lines examined while YAP KD only inhibits tube formation in MCF10A. For unknown mechanism, it is well known that YAP and TAZ have distinct and redundant functions in different tissues or/and phenotypes. It seems both the level and activity of YAP or TAZ are important in regulating a specific phenotype. Therefore, it is possible that although YAP and TAZ are paralogs TAZ may be more potent than YAP in regulating angiogenesis (i.e. activates more gene targets to mediate this function compared to YAP). As a result, in HUVECs and BOECs that express high levels of TAZ but low levels of YAP, knockdown of TAZ rather than YAP is sufficient to inhibit angiogenesis. However, in MCF10As that have higher levels of YAP but relatively low levels of TAZ, the angiogenesis we see is slightly more due to YAP function than TAZ such that YAP KD also significantly impairs angiogenesis. In addition, it was reported that YAP can negatively regulate TAZ in certain cell lines (Finch-Edmondson *et al*, 2015). Consistently, when YAP is knocked down, TAZ may be upregulated to compensate for the effect of YAP KD due to loss of negative regulation by YAP so that no phenotype is observed after YAP KD in some cell lines such as MDA-MB231 (Fig. 5j). Since MCF10A expresses high levels of YAP but low basal level of TAZ, YAP KD alone induces minimal TAZ activation for compensation (Fig. 5a) and, therefore, reduced angiogenesis is observed *in vitro* (Fig. 5c). Further more, while TAZ levels are low but retain high activity in these cells, the lack of compensation by YAP after its knockdown (TAZ cannot regulate YAP) may

explain why TAZ KD alone also causes reduced tube formation (Fig. 5c). While further studies to sort out this phenomenon are interesting, they are beyond the scope of this manuscript.

Page 9 – “While knockdown of both YAP and TAZ reduced both ANG-2 and CYR61 in MCF10A-VEGFR2 cells, TAZ knockdown alone was sufficient for loss of ANG-2 and CYR61 in HUVEC, BOEC, and MDA-MB-231 cells (Fig. 5a, 5d, 5g, 5j).” I do not understand these statements, and who overexpression was used for some experiments.

The purpose of using VEGFR overexpression in MCF10A was to help us explore angiogenesis that is specifically due to VEGFR signaling rather than other pathways which can play a role in this process. Since it could be argued that the angiogenesis we observed in MDA-MB-231, HUVEC and BOEC may be the result of multiple signaling pathways, we used a cell line that had very low basal angiogenesis (MCF10A) but that could be stimulated to form tubes with VEGFR overexpression and VEGF treatment. We have revised our statements accordingly to address these concerns.

The authors do not to explore whether VEGFR could be inhibiting a protein upstream of LATS1/2 (eg. MST1/2) – the inhibition of LATS1/2 could merely be a knock-on effect MST1/2 inhibition.

Whether the relationship between VEGFR and LATS is MST-dependent or –independent is an interesting area for further exploration. To examine this, we have stimulated VEGFR signaling by VEGF treatment in HUVEC cells and looked at phosphorylation of the components of our proposed signaling axis. VEGF induced phosphorylation of VEGFR, AKT and ERK. Interestingly, VEGF treatment also decreased phosphorylation (activation) of MST1 and LATS1 (at the site where MST1 phosphorylates LATS1, T1079) (Fig. 4p). Thus, MST likely plays a role in linking VEGFR with LATS inhibition. This is consistent with the previous literature showing that MST links the Hippo signaling pathway with AKT and MAPK signalling (Romano *et al.*, NCB, 2014). We have included this new data in our revised manuscript.

The authors state: ” Thus, ANG-2 and CYR61 may be critical downstream gene targets of YAP/TAZ mediating angiogenesis and vasculogenic mimicry (Fig. 5m). “ This is a possibility, but it is also likely that there are many other YAP/TAZ targets with altered expression levels downstream of VEGFR signalling that additionally, or alternatively, play a role in either endothelial or cancer cells. There is no evidence provided to argue that ANG-2 and Cyr61 are the specific hippo effectors regulating angiogenesis *in vitro* or *in vivo*. The authors could express ANG-2 or CYR61 to see if this would rescue the reduced tube formation seen in YAP/TAZ KD cells and determine whether formation is reduced if ANG-2 and/or CYR61 are knocked down. However, these *in vitro* assays would only raise the possibility of a role in angiogenesis.

We have performed the suggested experiment. As shown in Fig. 5m, 5n, and Supplementary Fig. 4k, administration of recombinant ANG2 and/or CYR61 can partially rescue *in vitro* tube formation indicating that these factors play some role in the process. However, since they cannot fully rescue tube formation, it is very possible that additional

factors are involved (as the reviewer describes). We have our statements accordingly.

Although the authors mention MST1/2 at the beginning of the paper, there has been no mention of this, or other proteins further upstream in the hippo pathway, after that. How can they be sure that VEGFR inhibits LATS and not MST1/2 etc, which then inhibit LATS as a knock-on effect?

As described above, we have added new data investigating the role of MST in our proposed signaling axis (see Fig. 4p). We have also added MST into our model (Fig. 7).

Figure 4: Contrary to what is said in the figure legend, data are represented as percentages in nucleus/cytoplasm/both, not as mean. How many independent experiments were carried out (as opposed to cells analysed or duplicates etc).

We thank the reviewer for identifying this error. The figure legend as well as the Materials and Methods section have been corrected. This data is the result of three independent experiments in which at least 200 cells were counted.

The authors should mention: Phosphorylation of angiotensin II by Lats1/2 kinases inhibits F-actin binding, cell migration, and angiogenesis. Dai X, She P, Chi F, Feng Y, Liu H, Jin D, Zhao Y, Guo X, Jiang D, Guan KL, Zhong TP, Zhao B. J Biol Chem. 2013 Nov 22;288(47):34041-51. doi: 10.1074/jbc.M113.518019. Epub 2013 Oct 8. PMID: 24106267

We have added this paper to our citations in the text.

Reviewer #3:

The manuscript by Azad and colleagues from Xiaolong Yang's laboratory reports on VEGFR as a possible upstream receptor that modulates Hippo-YAP pathway. The authors used a cleverly-designed LATS biosensor to monitor the effect of kinase inhibitors on the phosphorylation of YAP on Serine 127 residue that if it is decorated by phosphate serves generally as the inactivation signal for YAP protein. The work is novel and interesting; the Hippo-YAP pathway is being connected the angiogenic propensity of tumor formation.

We thank the reviewer for acknowledging the cleverly designed approach to designing our LATS biosensor, and the novel and interesting aspects of our research.

The following changes are suggested to improve the manuscript:

1. It would be important to monitor the biosensor fusion protein itself for changes in tyrosine phosphorylation status to eliminate a possibility that the basic constitutive phosphorylation of the biosensor is changed, for example, by inhibiting VEGFR or EGFR.

This is an interesting possibility! Thank you for the comment. To investigate this, we have looked at LATS-BS tyrosine phosphorylation in the cells used for our screen. As shown in Supplementary Fig. 1e, we see no basal phosphorylation of our LATS-BS. Furthermore, overexpression of VEGFR does not induce tyrosine phosphorylation of the LATS-BS. Therefore, we conclude it is unlikely that VEGFR affects LATS-BS activity by this mechanism.

2. It seems that there is “a back box” in terms of the presented data on the actual mechanism by which inhibition of VEGFR activates LATS kinase. Therefore it is suggested to be more careful using statements like “a novel VEGFR-PI3K/MAPK-LATS-YAP/TAZ-Cyr61/CTGF signaling axis”, or a new signaling pathway.

We have performed an additional experiment to attempt to resolve this “black box”. To investigate the role of MST in this signaling pathway, we stimulated HUVEC with VEGF and measured phosphorylation of each component (VEGFR, AKT, ERK, MST, LATS). VEGF treatment stimulates activation of VEGFR, AKT, ERK and inhibits MST (as measured by its own phosphorylation as well as its phosphorylation of LATS at T1079) and LATS (Fig. 4p). We have updated our statements in the revised manuscript to better reflect our findings related to this signalling pathway.

3. Serine 128 of YAP is a target of NEMO-like kinase that overrides Serine 127 phosphorylation of YAP. This could be discussed briefly, if space allows.

Thank you. We indeed previously reported that NLK can override YAP-S127 phosphorylation by phosphorylating YAP-S128 (Hong *et al.*, 2015). We have now included this in our discussion.

4. Basu, Subham and colleagues reported in 2003 (in Mol. Cell 11, pages 11-23) that YAP could be directly phosphorylated by AKT kinase on Serine 127. The scenario of AKT-YAP signaling and its regulation by VEGFR inhibition could be verified in cell models, or at least discussed.

Thank you. Both our research and studies by others showed that LATS1/2 instead of Akt is the major kinase phosphorylating S127 on YAP (Zhao *et al.*, 2007; Hao *et al.*, 2008). However, previous studies indeed showed that Akt can regulate the Hippo pathway by negatively regulating MST1 (Yuan *et al.*, 2010). We have referred this paper in our revised manuscript.

5. For sure the report is exciting in terms of a novel functional connection. To make this communication lucid, I would suggest that the length of the text be shortened by 40% focusing on the most important findings and selecting additional figures from the main body of the report to be placed in Supplementary Data. The presented here data on the LATS-YAP sensor are intriguing. The LATS-YAP sensor should be a valuable reagent for further exploration by the Hippo-YAP research community. It would be good if it is deposited to ADDGENE company for fast distribution among researchers.

Thank you! We have revised our statements to accurately reflect the significance of our work. We also aim to make our LATS biosensor easily available to other researchers so that it may become a valuable research tool for future studies.

Reviewers' comments:

Reviewer #1 (Remarks to the Author):

As noted in my initial review, I think this paper makes a significant contribution and the experimental analysis is convincing. I am also satisfied with their response to the other reviewers' comments. Thus I strongly support publication in Nature Communications.

I have to add though that they didn't directly address my minor comment on citing earlier work, perhaps because they are unaware of the relevant studies.

They wrote on p 8 lines 171-174

"In addition to VEGFR, our kinase inhibitor screen revealed that inhibitors of MEK, PI3K and AKT/PKB activated the LATS-BS (Fig. 3c), suggesting that VEGFR activates MAPK and PI3K signaling to suppress LATS and activate YAP/TAZ. To test this, we transfected the LATS-BS or STBS reporter alone or together with VEGFR2 into HEK293A cells and treated the cells with inhibitors of VEGFR, PI3K, AKT/PKB or MEK. Similar to the VEGFR inhibitor, PI3K, AKT and MEK inhibitors all blocked both VEGFR2-induced inhibition of LATS (Fig. 4n) and activation of YAP/TAZ (Fig. 4o)."

Since the observation that PI3K, AKT, and MEK affect Hippo signaling have already been published by others, I think it would be appropriate here for the authors to cite the initial publications showing this, which to my knowledge are:

for PI3K

Fan, R., Kim, N.-G., & Gumbiner, B. M. (2013). Regulation of Hippo pathway by mitogenic growth factors via phosphoinositide 3-kinase and phosphoinositide-dependent kinase-1. *Proceedings of the National Academy of Sciences of the United States of America*, 110(7), 2569–2574.

for AKT

Straßburger, K., Tiebe, M., Pinna, F., Breuhahn, K., & Teleman, A. A. (2012). Insulin/IGF signaling drives cell proliferation in part via Yorkie/YAP. *Dev Biol*, 367(2), 187–196.

for MEK

Reddy, B. V. V. G., & Irvine, K. D. (2013). Regulation of Hippo Signaling by EGFR-MAPK Signaling through Ajuba Family Proteins. *Developmental Cell*, 24(5), 459–471.

Reviewer #2 (Remarks to the Author):

This is a revised manuscript that has addressed many, but not all concerns of this reviewer, and there are some new concerns arising from data added during revision.

One prior paper showing that the Hippo pathway controls angiogenesis has now been cited, as requested: Phosphorylation of angiominin by Lats1/2 kinases inhibits F-actin binding, cell migration, and angiogenesis by Dai et al in *J Biol Chem*. 2013.

However, further papers were published whilst this manuscript was under re-review and show via YAP/TAZ-ECKO that VEGF regulates hippo signalling and that YAP/TAZ controls angiogenesis.

1. Sakabe et al. (PNAS 2017): YAP/TAZ-CDC42 signaling regulates vascular tip cell migration.
2. Wang et al. (*Dev Cell*, 2017) YAP/TAZ Orchestrate VEGF Signaling during Developmental Angiogenesis
3. Kim et al. (*JCI*2017): YAP/TAZ regulates sprouting angiogenesis and vascular barrier maturation

Given the prior publications, the authors' statement that they "identify novel upstream regulators of the Hippo pathway including VEGFR" is incorrect. I have to reiterate, the novelty of this publication is therefore the presentation of a novel tool, not a fundamental new scientific

discovery, although it does support the work that just just been published..

An important control that is now included in the paper shows that defective tube formation after inhibiting hippo signalling can be rescued with Cyr61/ANG2 expression.

However, the authors have not addressed the following reviewer queries:

- Why is there a much higher reduction in LATS-BS signal in the LATS1/2 KO compared with MST1/2 KO? Is something else able to phosphorylate, and activate, LATS1/2 when MST1/2 is not present?

- Page 9 "We used the same approach to examine the role of YAP and TAZ during angiogenesis using human umbilical vein endothelial cells (HUVEC) and BOECs. In these cells, endogenous YAP expression was barely detectable while TAZ was highly expressed (Fig. 5d, g)." It would be important to investigate the relative YAP and TAZ expression levels in all of the cell types used in this study.

- Page 9 – "Consequently, TAZ knockdown had a more substantial effect on blocking angiogenesis than YAP knockdown in both HUVECs and BOECs." In contrast, they show that TAZ is barely detectable in MCF10A cells, whereas YAP is high. Yet, KD of each has the same effect on tube formation - why might this be?

- Page 9 "While knockdown of both YAP and TAZ reduced both ANG-2 and CYR61 in MCF10A-VEGFR2 cells, TAZ knockdown alone was sufficient for loss of ANG-2 and CYR61 in HUVEC, BOEC, and MDA-MB-231 cells (Fig. 5a, 5d, 5g, 5j)." I do not understand these statements, and why overexpression was used for some experiments.

Additionally, the revision has raised new concerns:

Tube formation rescue: Did the authors confirm knockdown of YAP/TAZ in the experiments with siYAP/TAZ and double transfection for CYR61/ANG2 rescue? What is the number of independent experiments?

Verteporfin: Were experiments carried out in the dark to prevent cytotoxicity, especially during cell culture and aorta experiments? Verteporfin is a photosensitiser that has cytotoxic effects, so any light exposure would potentially be harmful and unspecifically compromise cell growth. A cytotoxicity assay on cells treated with Verteporfin or transfected with the siYAP/TAZ would be required. Importantly, a recent study suggests that verteporfin-mediated effects on hippo signaling can be explained by light-induced damage and is therefore non-specific (Konstantinou et al., Sci Reports 2017). The verteporfin experiments should therefore be controlled to show that cells are still viable and responsive to pathways other than those mediated by hippo signalling. Moreover, the verteporfin effect should be rescued as done for the siRNA experiments, by expressing with ANG2 or Cyr61. If such controls cannot be provided, the verteporfin experiments should be removed.

For the matrigel plugs, is it 3 plugs in 1 mouse or 3 plugs in 3 different mice?

The specificity of the inhibitors used is not accurately described. For example, SU4312 also inhibits PDGFRs.

Reviewer #3 (Remarks to the Author):

The authors revised the report very well and I think that the work should be of wide interest to a large community of researchers interested in the Hippo-YAP pathway.

Response to Reviewers' comments:

Reviewer #1 (Remarks to the Author):

As noted in my initial review, I think this paper makes a significant contribution and the experimental analysis is convincing. I am also satisfied with their response to the other reviewers' comments. Thus I strongly support publication in Nature Communications.

I have to add though that they didn't directly address my minor comment on citing earlier work, perhaps because they are unaware of the relevant studies.

They wrote on p 8 lines 171-174

“In addition to VEGFR, our kinase inhibitor screen revealed that inhibitors of MEK, PI3K and AKT/PKB activated the LATS-BS (Fig. 3c), suggesting that VEGFR activates MAPK and PI3K signaling to suppress LATS and activate YAP/TAZ. To test this, we transfected the LATS-BS or STBS reporter alone or together with VEGFR2 into HEK293A cells and treated the cells with inhibitors of VEGFR, PI3K, AKT/PKB or MEK. Similar to the VEGFR inhibitor, PI3K, AKT and MEK inhibitors all blocked both VEGFR2-induced inhibition of LATS (Fig. 4n) and activation of YAP/TAZ (Fig. 4o).”

Since the observation that PI3K, AKT, and MEK affect Hippo signaling have already been published by others, I think it would be appropriate here for the authors to cite the initial publications showing this, which to my knowledge are:

for PI3K

Fan, R., Kim, N.-G., & Gumbiner, B. M. (2013). Regulation of Hippo pathway by mitogenic growth factors via phosphoinositide 3-kinase and phosphoinositide-dependent kinase-1. *Proceedings of the National Academy of Sciences of the United States of America*, 110(7), 2569–2574.

for AKT

Straßburger, K., Tiebe, M., Pinna, F., Breuhahn, K., & Teleman, A. A. (2012). Insulin/IGF signaling drives cell proliferation in part via Yorkie/YAP. *Dev Biol*, 367(2), 187–196.

for MEK

Reddy, B. V. V. G., & Irvine, K. D. (2013). Regulation of Hippo Signaling by EGFR-MAPK Signaling through Ajuba Family Proteins. *Developmental Cell*, 24(5), 459–471.

We thank the reviewer for highlighting these additional studies. We have now cited them in the revised manuscript.

Reviewer #2 (Remarks to the Author):

This is a revised manuscript that has addressed many, but not all concerns of this reviewer, and there are some new concerns arising from data added during revision.

Thank you. In revising our initial submission, we performed additional experiments investigating the relationship between VEGFR and Hippo signaling based on this reviewer's comments:

1. Aortic ring angiogenesis assay demonstrates that pharmacological inhibition of YAP/TAZ diminishes VEGF-induced sprouting angiogenesis *ex vivo* by aortic endothelial cells (Fig. 6a,b,c, Supplementary Fig. 5)
2. Matrigel gel plug experiments show that pharmacological or genetic inhibition of YAP/TAZ inhibits VEGF-induced angiogenesis *in vivo* (Fig. 6d,e)
3. Mouse retinal angiogenesis assay indicates that pharmacological inhibition of YAP/TAZ suppresses VEGF-induced physiological angiogenesis *in vivo* (Fig. 6f)
4. Recombinant ANG2 and/or CYR61 partially rescues *in vitro* tube formation by HUVEC (Fig. 5m,n, Supplementary Fig. 4k)
5. Western blot analysis demonstrating that VEGF enhances phosphorylation of AKT and ERK while suppressing phosphorylation of MST1, LATS1 and YAP (Fig. 4p)

We have also addressed several other concerns of this reviewer in the manuscript with clarifications or in our previous response to reviewer comments document.

One prior paper showing that the Hippo pathway controls angiogenesis has now been cited, as requested: Phosphorylation of angiominin by Lats1/2 kinases inhibits F-actin binding, cell migration, and angiogenesis by Dai et al in J Biol Chem. 2013. However, further papers were published whilst this manuscript was under re-review and show via YAP/TAZ-ECKO that VEGF regulates hippo signalling and that YAP/TAZ controls angiogenesis.

1. Sakabe et al. (PNAS 2017): YAP/TAZ-CDC42 signaling regulates vascular tip cell migration.
2. Wang et al. (Dev Cell, 2017) YAP/TAZ Orchestrate VEGF Signaling during Developmental Angiogenesis
3. Kim et al. (JCI2017): YAP/TAZ regulates sprouting angiogenesis and vascular barrier maturation

Given the prior publications, the authors' statement that they "identify novel upstream regulators of the Hippo pathway including VEGFR" is incorrect. I have to reiterate, the novelty of this publication is therefore the presentation of a novel tool, not a fundamental new scientific discovery, although it does support the work that just been published.

We thank the reviewer for referring to these recent publications all of which were published during revision of our manuscript (Aug.18-Nov.1, 2017). These studies strongly support our data indicating that VEGF is a regulator of YAP/TAZ and underscore the significance of these observations. However, as the reviewer suggests, we should and have added these publications in the discussion of our revised manuscript. Following is the summary of these three publications:

1. Sakabe et al. *PNAS*, Sept. 25, 2017: *YAP/TAZ-CDC42 signaling regulates vascular tip cell migration*. This paper focuses on the roles of YAP/TAZ in vascular cell migration but does not investigate whether VEGF/VEGFR regulates YAP/TAZ in this process.
2. Kim et al. *JCI*, Sept. 1, 2017: *YAP/TAZ regulates sprouting angiogenesis and vascular barrier maturation*. This paper studies how YAP/TAZ regulate sprouting angiogenesis however does not explore how VEGFR regulates YAP/TAZ in angiogenesis.

3. Wang et al. *Dev Cell*, August 31, 2017: *YAP/TAZ Orchestrate VEGF Signaling during Developmental Angiogenesis*. Consistent with our observations, this paper demonstrates that VEGF regulates YAP/TAZ activity in angiogenesis. However, the scope of this work is primarily limited to the roles of VEGF-YAP/TAZ in angiogenesis during vascular development using YAP/TAZ conditional knockout mice. Mechanistically, the authors of this study suggest that VEGF regulates YAP/TAZ through cytoskeletal modifications (rather than through negative regulation of MST, LATS) and does not investigate whether YAP/TAZ downstream targets mediate VEGF-induced angiogenesis.

We would like to highlight the strengths of our experimental approach and novel findings exploring the relationship between VEGFR, YAP/TAZ and angiogenesis relative to the aforementioned studies:

1. Using the LATS biosensor (LATS-BS) we determined that VEGFR modulates Hippo pathway activity and uncovered evidence supporting a VEGF-VEGFR-MAPK/PI3K-LATS-YAP/TAZ-ANG2/CYR61 signaling axis regulating angiogenesis (Fig. 4-7).
2. We have validated VEGFR-Hippo signaling through biochemical and functional experiments using multiple cell lines of different tissue types: human HEK293, immortalized mammary MCF10A cells, MDA-MB231 breast cancer cells, and endothelial cells (HUVEC and BOEC) (Fig. 4).
3. Using MDA-MB231, we have also shown that YAP and TAZ play important roles in VEGF-induced tumor vasculogenic mimicry (VM) (Fig. 5j-i).
4. We have shown that YAP/TAZ are critical for VEGF-induced angiogenesis rat aortic angiogenesis assays (Fig. 6a-c).
5. We have shown that genetic knockdown of YAP/TAZ or pharmacological inhibition of YAP/TAZ by VP prevents VEGF-induced angiogenesis *in vivo* through matrigel plug experiments (Fig. 6d-e).

Finally, we would like to discuss the areas of our study that do not overlap with the new literature. As previously described, the Hippo pathway has emerged as a key signaling pathway that regulates a wide range of biological processes (much like MAPK and PI3K signaling pathways). Therefore, identification of novel functions and regulators of the Hippo pathway is one of the hottest research areas in molecular and cell biology, immunology, developmental biology, and cancer biology (Leach *et al.*, *Nature*, 2017; Morikawa *et al.*, *Nature*, 2017; Britschgi *et al.*, *Nature*, 2017; Wang *et al.*, *Nat Immunol.*, 2017; Li *et al.*, *Nat Cell Biol.*, 2017; Zhang *et al.*, *Nat Cell Biol.*, 2017; Elosegui-Artola *et al.*, *Cell*, 2017; Hubaud *et al.*, *Cell*, 2017; Nakajima *et al.*, *Dev Cell*, 2017; Su *et al.*, *Dev Cell*, 2017). However, until now there has been no tool available to measure the dynamics and activity of Hippo signaling in a quantitative, real-time, high-throughput and non-invasive manner. We have developed and validated a novel bioluminescent LATS-BS (Fig. 1-2). As an illustration of the utility of the LATS-BS, we performed a small-scale kinase inhibitor screen and identified several regulators of LATS kinase activity that were previous unreported in the literature (e.g. VEGFR, TrkA, SYK, SGK, etc) (Fig. 3). We further validated the relationship between VEGFR and Hippo signaling in order to test whether our LATS-BS could uncover *bona fide* biological relationships with functional significance (Fig. 4-7). We anticipate that our LATS-BS will have numerous applications in future studies (e.g. for large-scale screens, translational studies) and, as the third reviewer states,

would “be of wide interest to a large community of researchers interested in the Hippo-YAP pathway” as well as to the diverse readers of *Nat Commun*. Indeed, several other similar biosensors have been reported in recent issues of *Nat Commun*:

1. Kono et al. *Nat Commun*. 8:1163 (2017): Bioluminescence imaging of G protein-coupled receptor activation in living mice.
2. Bertolin et al., *Nat Commun*. 7: 12672 (2016): A FRET biosensor reveals spatiotemporal activation and functions of aurora kinase A in living cells
3. Nadler et al. *Nature Commun*. 7, 12266 (2016): Rapid construction of metabolite biosensors using domain-insertion profiling.
4. Lima-Fernandes et al., *Nat. Commun*. 5:4431 (2014): A biosensor to monitor dynamic regulation and function of tumour suppressor PTEN in living cells.
5. Seong et al., *Nat. Commun*. 2:1593 (2011): Detection of focal adhesion kinase activation at membrane microdomains by fluorescence resonance energy transfer.

Given all of this, we feel that our study represents an advancement in our understanding of the Hippo pathway, will provide a strong foundation for future work and would be appropriate for publication in *Nature Commun*.

An important control that is now included in the paper shows that defective tube formation after inhibiting hippo signalling can be rescued with Cyr61/ANG2 expression.

However, the authors have not addressed the following reviewer queries:

- Why is there a much higher reduction in LATS-BS signal in the LATS1/2 KO compared with MST1/2 KO? Is something else able to phosphorylate, and activate, LATS1/2 when MST1/2 is not present?

As described in our previous responses to reviewers’ comments, there are other factors that can regulate LATS1/2 independent of MST1/2 (e.g. MAP4K, ubiquitin ligases ITCH and WWP1, Cdc2/Cyclin B, Aurora-A). Thus, in the absence of MST1/2 multiple factors may maintain/preserve LATS1/2 kinase activity. This explains why there is a greater reduction in LATS-BS signal when LATS1/2 is knocked-out compared to when MST1/2 is knocked-out. We have described this clearly in the revised manuscript.

- Page 9 “We used the same approach to examine the role of YAP and TAZ during angiogenesis using human umbilical vein endothelial cells (HUVEC) and BOECs. In these cells, endogenous YAP expression was barely detectable while TAZ was highly expressed (Fig. 5d, g).” It would be important to investigate the relative YAP and TAZ expression levels in all of the cell types used in this study.

As described in our previous responses to reviewers’ comments, our Western blot exposures were chosen to reflect the relative expression of YAP and TAZ in each of the cell lines. Thus, as shown in Fig. 5a,d,g,j, MCF10A is YAP (high) and TAZ (low); HUVEC/BOEC are YAP (low) and TAZ (high); MDA-MB-231 are YAP (high) and TAZ (high). We have stated this clearly in the figure legend of Fig. 5a & d-l in the manuscript.

- Page 9 – “Consequently, TAZ knockdown had a more substantial effect on blocking angiogenesis than YAP knockdown in both HUVECs and BOECs.” In contrast, they show that TAZ is barely detectable in MCF10A cells, whereas YAP is high. Yet, KD of each has the same effect on tube formation - why might this be?

In our previous responses to reviewers’ comments, we describe several mechanisms that might explain this observation (see below). Since these are only our observations, we have not included extensive discussion related to this in the manuscript.

To address the reviewer’s question, these data are our observations, which we interpret to mean that YAP/TAZ double knockdown inhibits tube formation. We currently do not know why single knockdown of TAZ inhibits tube formation in all of the cell lines examined while single YAP KD inhibits tube formation only in MCF10A. However, there are several factors which might be contributing to these observations. First, it is well appreciated that YAP and TAZ have distinct and redundant functions in different tissues and/or phenotypes, although the mechanisms underlying this are currently unknown. Thus, it is possible that TAZ is compensating for YAP KD in certain cell lines. Second, although YAP and TAZ are paralogs TAZ may be more potent than YAP in regulating angiogenesis (i.e. activates more gene targets to mediate this function compared to YAP). As a result, in HUVECs and BOECs that express high levels of TAZ but low levels of YAP, knockdown of TAZ rather than YAP is sufficient to inhibit angiogenesis. However, in MCF10As that have higher levels of YAP but relatively low levels of TAZ, the angiogenesis we see may be slightly more due to YAP function than TAZ such that YAP KD also significantly impairs angiogenesis. Finally, to further complicate the model, it was reported that YAP can negatively regulate TAZ in certain cell lines (Finch-Edmondson *et al*, 2015). So it is possible that when YAP is knocked down, TAZ may be upregulated such that no phenotype is observed after YAP KD in some cell lines such as MDA-MB231 (Fig. 5j).

- Page 9 “While knockdown of both YAP and TAZ reduced both ANG-2 and CYR61 in MCF10A-VEGFR2 cells, TAZ knockdown alone was sufficient for loss of ANG-2 and CYR61 in HUVEC, BOEC, and MDA-MB-231 cells (Fig. 5a, 5d, 5g, 5j).” I do not understand these statements, and why overexpression was used for some experiments.

We have already revised these statements in our previous revision so that they are more clear.

Additionally, the revision has raised new concerns:

Tube formation rescue: Did the authors confirm knockdown of YAP/TAZ in the experiments with siYAP/TAZ and double transfection for CYR61/ANG2 rescue? What is the number of independent experiments?

Indeed, we confirmed knockdown of YAP/TAZ in the CYR61/ANG2 tube formation rescue experiments (see below). Recombinant CYR61/ANG2 was purchased and used to rescue the siYAP/TAZ phenotype rather than cDNA transfection. This experiment was repeated twice with 3 biological replicates per treatment condition.

Replicate 1:

Replicate 2:

Verteporfin: Were experiments carried out in the dark to prevent cytotoxicity, especially during cell culture and aorta experiments? Verteporfin is a photosensitizer that has cytotoxic effects, so any light exposure would potentially be harmful and unspecifically compromise cell growth. A cytotoxicity assay on cells treated with Verteporfin or transfected with the siYAP/TAZ would be required. Importantly, a recent study suggests that verteporfin-mediated effects on hippo signaling can be explained by light-induced damage and is therefore non-specific (Konstantinou et al., Sci Reports 2017). The verteporfin experiments should therefore be controlled to show that cells are still viable and responsive to pathways other than those mediated by hippo signalling. Moreover, the verteporfin effect should be rescued as done for the siRNA experiments, by expressing with ANG2 or Cyr61. If such controls cannot be provided, the verteporfin experiments should be removed.

Verteporfin (VP) was originally identified as a drug that directly inhibits YAP activity by disrupting YAP-TEAD interaction in cells (Liu-Chittenden et al., 2012). Since its discovery, it has become the most commonly used inhibitor of YAP/TAZ activity *in vitro* and *in vivo* (43 papers published so far; see most recent publication using VP in Cosset et al., *Cancer Cell*, Dec.11 2017). However, like many other drugs, high doses of VP may have non-specific effects. To account for this, we have only used low concentrations (50-200 nM) of VP in all of our experiments. Furthermore, as the reviewer describes, VP is a photosensitive molecule. Therefore, appropriate precautions were taken throughout our experimental procedure in order to limit light exposure. During VP treatment, cells were cultured in the dark. Rat aortic angiogenesis assays were also performed in the dark.

We have previously explored the possibility for cytotoxicity due to YAP/TAZ genetic knockdown in our initial submission and showed that knockdown of YAP/TAZ does not cause cell death (see Supplementary Fig. 4f). Nonetheless, in response to this reviewer's concerns, we have performed additional experiments investigating VP:

1. VP does not cause apoptosis in HUVEC cells as demonstrated by cleaved PARP (Supplementary Fig. 5d).

2. Recombinant administration of ANG2 and CYR61 partially rescues the effect of VP on VEGF-induced angiogenesis *in vitro* (Supplementary Fig. 5b,c).

We have added this new data into our revised manuscript.

For the matrigel plugs, is it 3 plugs in 1 mouse or 3 plugs in 3 different mice?

For the matrigel plug experiments, two plugs were implanted per mouse. In designing this experiment, we placed two different plugs per mouse (e.g. siCtrl and siYAP/TAZ in the same mouse for the HUVEC experiments) so that each experimental condition was represented across multiple, different mice. Thus, 3 plugs across 3 different mice were used per condition. We have clarified this in the methods of our revised manuscript.

The specificity of the inhibitors used is not accurately described. For example, SU4312 also inhibits PDGFRs.

In describing the inhibitors used in this screen, we considered the concentration of drug administered (10 μ M) as well as the IC₅₀ for each target of the inhibitor. For example, SU4312 inhibits VEGFR and PDGFR with IC₅₀ of 0.8 μ M and 19.4 μ M, respectively (25-fold difference). Thus, we described SU4312 as a VEGFR inhibitor in our manuscript. To account for potential off-target effects of SU4312, we also used other inhibitors of VEGFR function (apatinib, axitinib) (Fig. 4a,b,c). We have amended the figure legend of Fig. 3c,d to clarify that the targets listed are the primary targets of each drug at 10 μ M concentration.

Reviewer #3 (Remarks to the Author):

The authors revised the report very well and I think that the work should be of wide interest to a large community of researchers interested in the Hippo-YAP pathway.

We thank the reviewer for recognizing the significant contribution our study will make to the field.